# On The Landscape of Spoken Language Models:
# A Comprehensive Survey

**Siddhant Arora**[1*] **Kai-Wei Chang**[2*] **Chung-Ming Chien**[3*] **Yifan Peng**[1*] **Haibin Wu**[2*#]
**Yossi Adi**[4+] **Emmanuel Dupoux**[5+] **Hung-Yi Lee**[2+] **Karen Livescu**[3+] **Shinji Watanabe**[1+]
[1] **Carnegie Mellon University, USA**
[2] **National Taiwan University, Taiwan**
[3] **Toyota Technological Institute at Chicago, USA**
[4] **Hebrew University of Jerusalem, Israel**
[5] **ENS - PSL, EHESS, CNRS, France**

**Reviewed on OpenReview:** `https://openreview.net/forum?id=BvxaP3sVbA`

## Abstract

The field of spoken language processing is undergoing a shift from training custom-built, task-specific models toward using and optimizing *spoken language models* (SLMs) which act as universal speech processing systems. This trend is similar to the progression toward universal language models that has taken place in the field of (text) natural language processing. SLMs include both "pure" language models of speech—models of the distribution of tokenized speech sequences—and models that combine speech encoders with text language models, often including both spoken and written input or output. Work in this area is very diverse, with a range of terminology and evaluation settings. This paper aims to contribute an improved understanding of SLMs via a unifying literature survey of recent work in the context of the evolution of the field. Our survey categorizes the work in this area by model architecture, training, and evaluation choices, and describes some key challenges and directions for future work.

## 1 Introduction

In the last few years, the field of natural language processing (NLP) has evolved from (1) training many task-specific models from scratch, to (2) combining pre-trained multi-purpose contextual representation models (such as BERT (Devlin et al., 2019)) with a small number of task-specific parameters, to (3) training generative *universal* large language models (LLMs) (Brown et al., 2020; OpenAI et al., 2024)[1] that perform arbitrary text tasks given natural language instructions (prompts) and can generalize to unseen domains and tasks (Wei et al., 2022a; Liu et al., 2023), and finally to (4) dialogue (chatbot) systems that function as assistants and perform tasks while directly interacting with the user.

The field of speech processing has been undergoing a similar evolution, although with some lag, and has mainly focussed on stages (1) and (2). The current state of the art (SOTA) for common specialized speech tasks—including automatic speech recognition (ASR), speech translation (ST), spoken language understanding (SLU) tasks, and speaker identification (SID)—involves combining a pre-trained self-supervised encoder (Mohamed et al., 2022) with a task-specific prediction "head". For some very common tasks—namely, ASR and ST—in relatively high-resource languages, large supervised models (Radford et al., 2023; Peng et al., 2023b) also have consistently good (if not SOTA) performance.

Recent work has begun to develop *spoken language models* (SLMs), analogous to text LLMs, which are in principle capable of performing arbitrary speech tasks given natural language instructions. However, the

---

[1]Throughout the paper, we use the terms LLMs and LMs interchangeably to refer to modern language models.

Table 1: Typology of text and spoken LMs. We use a loose notation here, where *speech* and *text* are to be interpreted in context; for example, $p(text|text)$ in post-trained text LMs corresponds to modeling some desired output text given an input text instruction or prompt. "Post-training" refers to any form of instruction-tuning and/or preference-based optimization. Please see the sections below for details and references for the example models.

| Type of LM | Training Strategy | Model distribution | Examples |
|---|---|---|---|
| pure text LM | pre-training | $p(text)$ | GPT, LLaMA |
| pure text LM | post-training | $p(text|text)$ | ChatGPT, LLaMA-Instruct |
| pure speech LM | pre-training | $p(speech)$ | GSLM, AudioLM, TWIST |
| pure speech LM | post-training | $p(speech|speech)$ | Align-SLM |
| speech+text LM | pre-training | $p(text, speech)$ | SpiRit-LM, Moshi (pre-trained) |
| speech+text LM | post-training | $p(text, speech|text, speech)$ | Moshi (post-trained), Mini-Omni |
| speech-aware text LM | post-training | $p(text|speech, text)$ | SALMONN, Qwen-Audio-Chat |

term "SLM" has not been standardized in the literature and has been used to refer to a wider range of model types than text language models. Several common classes of models have emerged, all referred to as SLMs: (1) *pure SLMs*: models of the distribution of speech, $p(speech)$, typically trained on unlabeled tokenized speech data only with a next-token prediction objective (similarly to the pre-training phase in text LLMs) (Lakhotia et al., 2021); (2) *speech+text SLMs*: models of the joint distribution of speech and the corresponding text $p(text, speech)$, typically trained using paired (speech, text transcription) data, which can be viewed as a direct extension of class (1) (Nguyen et al., 2025); and (3) *speech-aware text LMs*: models that combine text LLMs with pre-trained speech encoders to retain the instruction-following (IF) characteristics of the text LLMs and reason about the input speech or audio recordings (Tang et al., 2024; Chu et al., 2023). Approaches in category (3) model a conditional distribution, $p(text|speech, text)$, of desired output text in response to the input speech and text instruction.

Following the text LLMs analogy, models in categories (1) and (2) can be viewed as analogues to pre-trained LLMs. Like text LLMs, these models can be post-trained—via instructions, preferences, or other means—to learn the distributions of desired output speech (for category (1)) or output speech+text (category (2)) given desired inputs (speech or speech+text, respectively). Models in category (3) typically start with fine-tuned LLMs, but involve some additional post-training to both align the speech and text representations and enable the model to perform new speech-specific tasks. Table 1 provides a typology of these categories along with example models from the recent literature. Figure 1 presents a timeline of additional models and their development, and Tables 4 and 5 in the Appendix provide additional model details.

Although existing models have been applied to different tasks using different priors and modeling techniques, all of the three categories of SLMs mentioned above form steps on the way to *universal speech processing systems*. For the purposes of this paper, we define a universal speech processing system as a model that satisfies the following criteria:

1. It has both spoken input and spoken output with optional text input and/or output. The spoken input may serve as either an instruction or a context.

2. It is intended to be "universal"; that is, it should in principle be able to address arbitrary spoken language tasks, including both traditional tasks and more complex reasoning about spoken data.

3. It takes instructions or prompts in the form of natural language (either speech or text), and not, for example, only task specifiers (Radford et al., 2023) or soft prompts (Chang et al., 2024).

This is a functional definition, and does not restrict the form of the model. We also note that most of the models in the current literature, and therefore in this survey, do not satisfy all of these criteria; for example, many do not have speech as both input and output, and many are trained and evaluated on a fairly limited set of tasks. However, the models we include can all be seen as steps toward the same goal of eventually

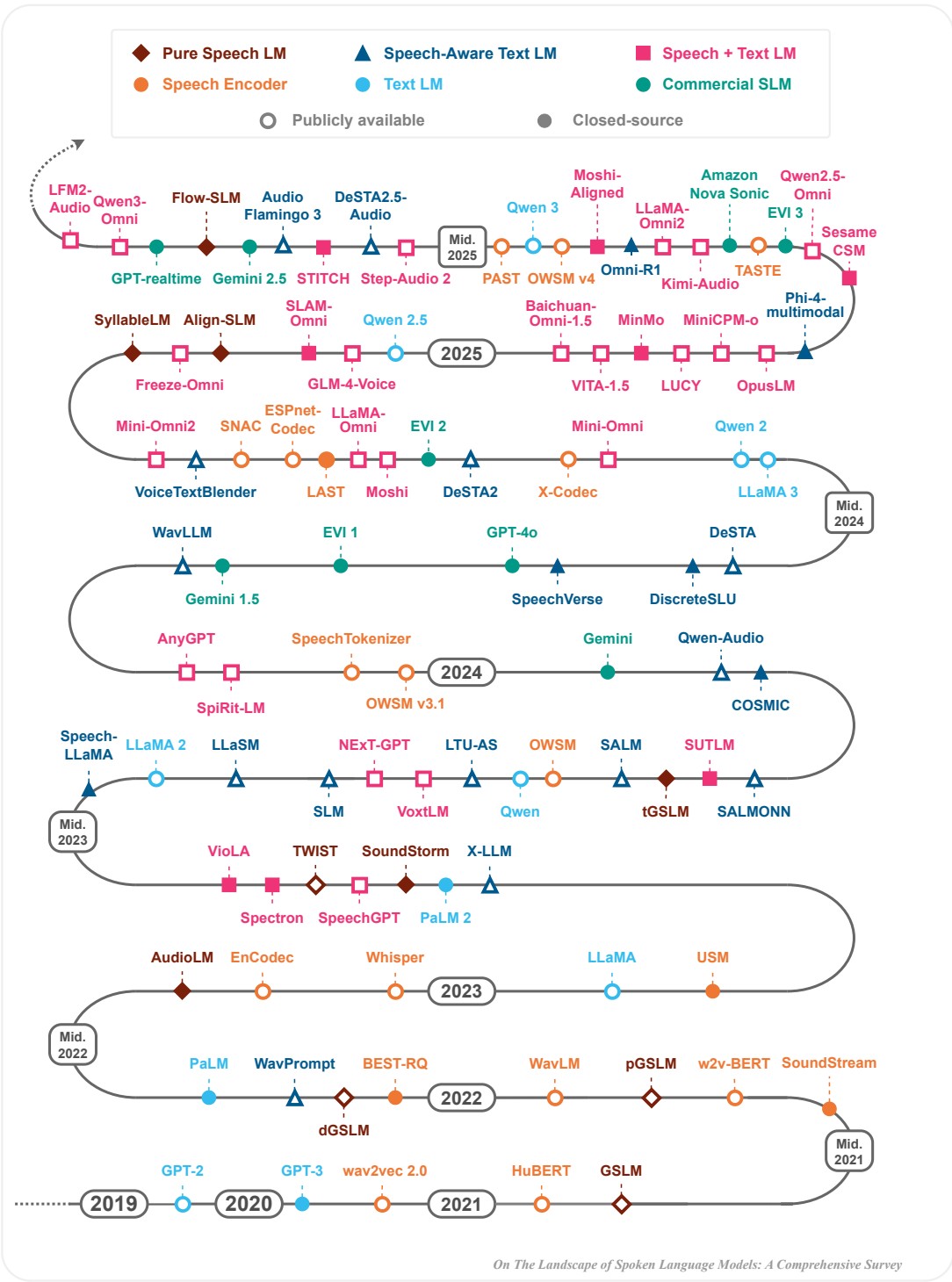

Figure 1: Development timeline of spoken language models. "Publicly available" refers to models with publicly released weights (but not necessarily code, data, or other artifacts). "Commercial SLMs" include some systems that handle additional (non-linguistic) modalities, such as images. For more details and references, see the main article and Tables 4 and 5.

developing SLMs that can serve as universal speech processing systems, in the same way that pre-trained and post-trained text LLMs have served as steps toward universal written language processing systems.

One may wonder whether a better path to universal speech processing is to combine speech recognition, text LLMs, and speech synthesis in series (often referred to as a "cascade"). This is indeed a strong baseline approach for many tasks (Huang et al., 2025; Yang et al., 2024b). However, some tasks require access to aspects of the audio signal beyond the word string, such as properties of the speaker, their emotional state, prosody, or other acoustic properties. For example, detecting sarcasm often involves aspects of prosody that indicate the speaker's meaning may be the opposite of the naive interpretation of the uttered word sequence. In addition, even for tasks that are in principle addressable using the word string alone, an end-to-end SLM approach can avoid the compounding of errors that can occur when combining ASR, text LLMs, and TTS systems in series. Similar observations have been made in earlier work, in the context of spoken language understanding via cascaded ASR and NLP models versus end-to-end models (Haghani et al., 2018; Chen et al., 2018). Finally, even when a cascade approach performs well, it can introduce substantial latency, because it requires several systems to be run in series although only the output of the final one is needed.

The past few years have seen a major acceleration in work on SLMs. However, there have been limited efforts to survey the design and modeling choices made in this work. Additionally, these models are often evaluated on very different tasks and datasets, making it difficult to assess the relative performance of different approaches. As part of this survey, we collect and organize many of the existing evaluations (though standardized evaluation is still one of the remaining challenges in this research area). Lastly, as SLMs have been viewed and defined differently depending on the intended applications and modeling choices, we provide a unified formulation and terminology for describing SLMs (Section 2).

This survey is intended to serve as a snapshot of the current moment in the evolution of the field, providing a review of the recent literature and a unified definition of SLMs and their components. While new SLMs are being proposed at a steady pace, we hope that this survey of the research landscape will help readers more easily place new models in context. We aim to provide an improved understanding of the successes and remaining limitations of SLMs developed thus far, along the path to SLMs as universal speech processing systems.

**Scope of this survey**   Although the ultimate goal is task-independent models that can be instructed with natural language, many LM-inspired approaches thus far have been task-specific (for example, special cases of conditional $p(text|speech)$ models for speech recognition and translation such as Whisper (Radford et al., 2023) and OWSM (Peng et al., 2023b), and conditional $p(speech|text)$ models for text-to-speech synthesis such as VALL-E (Chen et al., 2025b) and VoiceBox (Le et al., 2024)), and some have been more general but rely on task tokens or soft prompts (e.g., Qwen-Audio (Chu et al., 2023) and SpeechPrompt (Chang et al., 2024) respectively). In this survey, we focus on models that are at least in principle task-independent (although they may have been tuned on a relatively small set of tasks) and that take natural language as input rather than task tokens or soft prompts. We will, however, discuss some of the important task-specific models as they relate to the more general models.

In addition, as discussed in Section 2, SLMs often comprise several components including a speech encoder, speech decoder, speech-text alignment module (when applicable), and sequence model. In this survey, we provide relatively brief descriptions of speech encoding and decoding; these have been covered well in previous surveys (e.g., Wu et al. (2024a)), and our main focus is on the aspects that are specific to SLMs, such as sequence modeling and speech-text alignment.

Finally, music and general audio share many properties with speech. A number of language modeling approaches have been developed specifically for music and general audio (e.g., (Copet et al., 2023; Agostinelli et al., 2023)), and some SLM research combines speech and other audio (e.g., (Gong et al., 2023b)). In this survey we include non-speech audio only to the extent that it is used in the context of SLMs.

**Related surveys**   To place this paper in context, we note that several other recent surveys have addressed aspects of SLMs. Zhang et al. (2024a) and Chen et al. (2024a) review multi-modal LLMs, including a limited subset of SLMs. By focusing on SLMs as our main subject, we survey a substantially larger set of

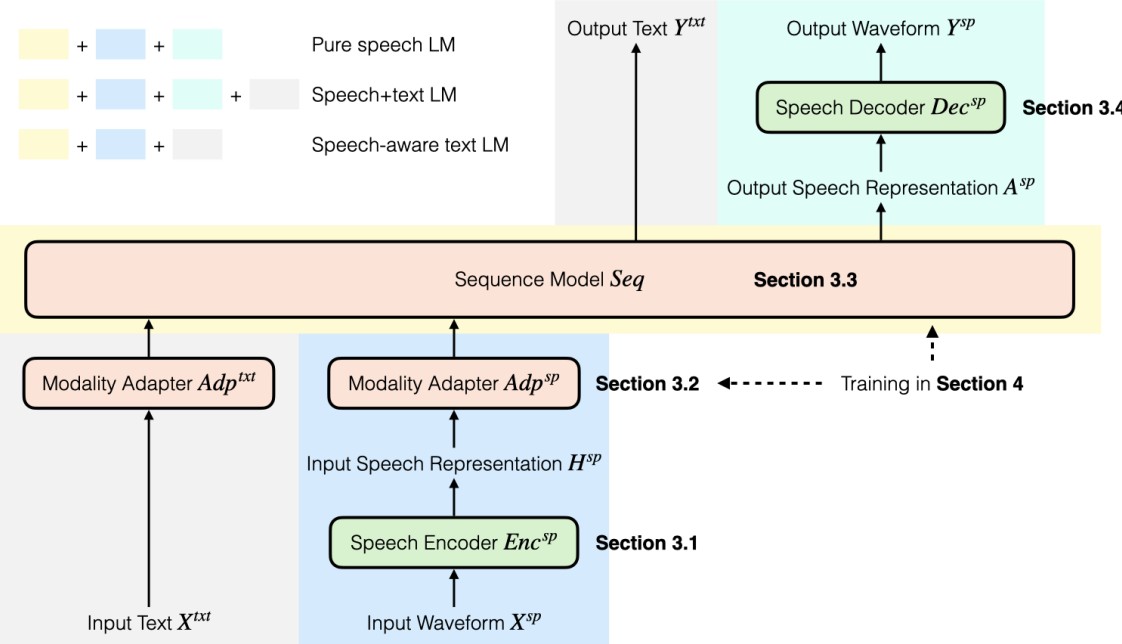

Figure 2: Overview of SLM architecture. See Sections 3 and 4 for more detailed descriptions of the components and training methods, respectively.

models in greater detail and explore speech-specific issues. Latif et al. (2023) provide a survey on large audio models. This previous survey covers a much larger space of audio language models (including environmental sounds, music, and other audio) and therefore does not present SLMs in as much detail, nor include recent *instruction-following* (IF) models from the past year. Another recent survey (Wu et al., 2024a) provides an overview of neural audio codecs and codec-based (Section 3) models, which mainly focuses on speech tokenization rather than SLMs more broadly. Guo et al. (2025) and Mousavi et al. (2025) also survey tokenization methods. Ji et al. (2024) provide a survey of SLMs, but their main focus is on spoken dialogue systems, rather than the full range of SLMs covered here. Another related SLM survey paper (Peng et al., 2024a) focuses on speech-aware text LMs but leaves out other types of SLMs. Finally, concurrent work by Cui et al. (2025) surveys a similar range of SLMs; we provide a complementary view of this research area, covering certain aspects in greater detail and summarizing others more briefly. Our main goal is to provide a unifying view that shows the connections and distinctions between various SLM approaches and helps the reader navigate the changing research landscape.

**Outline** In the remaining sections, we begin by outlining a general formulation of SLMs (Section 2), followed by a discussion of various design choices (Section 3), including speech encoders and decoders, modality adapters, and sequence modeling. We then describe the multiple optimization pipelines used for training SLMs (Section 4). In Section 5, we review and categorize some of the notable recent models. In Section 6, we discuss how SLMs have been adapted for dialogue ("full-duplex") mode. In Section 7, we present existing approaches for evaluating SLMs. Finally, we conclude by discussing the limitations of current approaches and provide some suggestions for future research (Section 8).

## 2 Overall architecture

We start by providing a *general* formulation (shown in Figure 2) of SLMs, which takes either/both speech and text as input and generates either/both speech and text (where at least one of the input or output

includes speech). Although SLMs differ in many of their design choices, this unifying description subsumes all of the models we cover.[2]

Let $X^{\text{txt}} \in \mathcal{V}^*$ and $X^{\text{sp}} \in \mathbb{R}^*$ denote the input text and speech, respectively. That is,

$$X^{\text{txt}} = \{t_1, t_2, \ldots, t_N\} \tag{1}$$

is a sequence of text tokens $t_i$ of arbitrary length $N$ drawn from a vocabulary $\mathcal{V}$. The speech input is a waveform, that is a sequence

$$X^{\text{sp}} = \{s_1, s_2, \ldots, s_T\} \tag{2}$$

of real-valued samples of arbitrary length $T$, where typically $T \gg N$. We first map $X^{\text{sp}}$ using the speech encoder $\text{Enc}^{\text{sp}}()$ to a sequence of speech representations $H^{\text{sp}}$:

$$H^{\text{sp}} = \text{Enc}^{\text{sp}}(X^{\text{sp}}). \tag{3}$$

The resulting speech representations of arbitrary length $L$, $H^{\text{sp}} = \{h_1, h_2, \ldots, h_L\}$, can take one of two forms, either a sequence of $d$-dimensional continuous vectors $h_l \in \mathbb{R}^d$ or a sequence of discrete tokens $h_l \in \mathcal{D}$ drawn from a learned codebook $\mathcal{D}$.

The speech representations $H^{\text{sp}}$ are further transformed through a modality adapter

$$\text{Adp}^{\text{sp}}(H^{\text{sp}}) \in \mathbb{R}^{L' \times d'}, \tag{4}$$

where typically $L' \leq L$ and $d'$ is the dimension of the transformed representation. This transformation is intended to improve the alignment of the speech representations with the sequence model, which may have been pre-trained as a text model. In addition, the length disparity between speech and text sequences can be addressed at this stage by applying temporal adjustments in $\text{Adp}^{\text{sp}}()$. The text token sequence $X^{\text{txt}}$ is also transformed into a sequence of vector representations, with the same dimensionality $d'$, via an adapter

$$\text{Adp}^{\text{txt}}(X^{\text{txt}}) \in \mathbb{R}^{N' \times d'}. \tag{5}$$

The text adapter is typically simply an embedding table, and therefore $N' = N$.[3] After the adapters, therefore, the text and speech are mapped to the same representation space with dimension $d'$, but the text and speech representation sequences are still not necessarily the same length (i.e., $L'$ and $N'$ need not be identical).

Next, given the adapted input text and/or speech representations, a sequence model, denoted $\text{Seq}()$, generates outputs, typically in an autoregressive manner. Here, we assume that each invocation of $\text{Seq}()$ corresponds to one generation step. For the model types described in Table 1, the formulations are as follows:

**Pure speech LMs**   The output of $\text{Seq}()$ is a speech representation

$$h' = \text{Seq}(\text{Adp}^{\text{sp}}(H^{\text{sp}})). \tag{6}$$

The input to $\text{Seq}()$ is a sequence of representations of the generated speech so far, with shape $L' \times d'$, and the output speech representation $h'$ is either a continuous vector (in $\mathbb{R}^d$) or a discrete token (in $\mathcal{D}$). Following the usual autoregressive generation formulation, $h'$ is then appended to $H^{\text{sp}}$, increasing its length by one, $\text{Seq}()$ then generates the next output, and so on. All of the generated outputs $h'$ together form a sequence $A^{\text{sp}} = \{h'_1, h'_2, \ldots, h'_{L''}\}$ with sequence length $L''$. Finally, the speech decoder $\text{Dec}^{\text{sp}}$ converts the output speech representations $A^{\text{sp}}$ to a waveform $Y^{\text{sp}}$:

$$Y^{\text{sp}} = \text{Dec}^{\text{sp}}(A^{\text{sp}}). \tag{7}$$

---

[2]We do make some assumptions, for example, that the sequence model is autoregressive, which in principle need not hold but in practice do hold for current models.

[3]We are not aware of any SLMs that use any text adapters besides the typical embedding table, nor models where $N' \neq N$, but we allow this possibility in the definition for maximal generality.

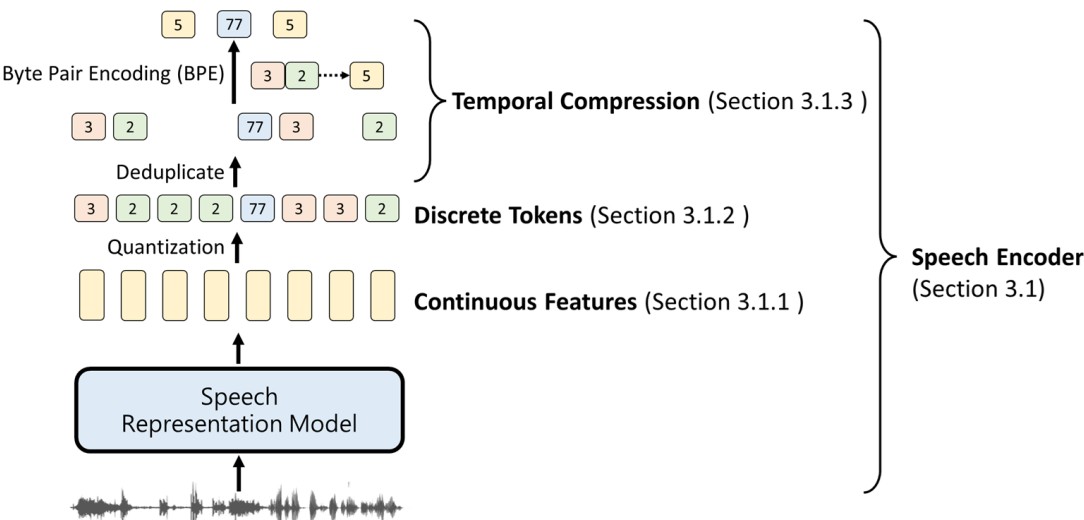

Figure 3: A general pipeline for speech encoders. Note that different encoders use different components of the pipeline. See Section 3.1 for more details.

**Speech-aware text LMs** In this type of model, the sequence model Seq() generates a text token $t'$ according to

$$t' = \text{Seq}\Big(\text{Adp}^{\text{sp}}(H^{\text{sp}}),\ \text{Adp}^{\text{txt}}(X^{\text{txt}})\Big) \tag{8}$$

Here, the input to Seq() is a concatenation of the adapted speech representation $\text{Adp}^{\text{sp}}(H^{\text{sp}})$ and text representation $\text{Adp}^{\text{txt}}(X^{\text{txt}})$, forming a tensor with shape $(L' + N') \times d'$. The output is a text token $t' \in \mathcal{V}$. This generated token is then appended to $X^{\text{txt}}$, and the process is repeated to generate subsequent tokens. Finally, the sequence of all generated tokens forms the output text sequence $Y^{\text{txt}}$.

**Speech+text LMs** In this more complex scenario, speech and text are modeled jointly:

$$h', t' = \text{Seq}(\text{Adp}^{\text{sp}}(H^{\text{sp}}), \text{Adp}^{\text{txt}}(X^{\text{txt}})). \tag{9}$$

Here both the inputs and outputs are "hybrid" representations consisting of a combination of speech and text representations. There are multiple approaches for generating such hybrid speech+text representations, which will be discussed in Section 3.3.2.

The encoder, decoder, and sequence model Seq() are often pre-trained separately. The modality adapter is typically trained from scratch, sometimes along with fine-tuning of the sequence model. The encoder and decoder parameters are usually fixed after pre-training. Section 3 provides more detailed information about the model components, while Section 4 describes typical training methods.

## 3 SLM components

### 3.1 Speech Encoder

In text LMs, text is typically tokenized into subword units, generated through methods like byte pair encoding (BPE) (Gage, 1994), and these tokens are then represented as vector embeddings. In contrast, speech is a continuous waveform with no obvious tokenization and no separation between linguistic information and other acoustic aspects. The task of the speech encoder, shown in Figure 3, is to extract meaningful representations from the waveform.

The first part of the encoder is a speech representation model, which transforms the speech signal into continuous features (vectors). These continuous features can either be used directly as input to the speech

modality adapter $\text{Adp}^{\text{sp}}(\cdot)$ or quantized into discrete units (using, e.g., $k$-means or VQ-VAE (van den Oord et al., 2017)). In general, pure speech LMs and speech+text LMs tend to use discrete speech tokens, whereas speech-aware text LMs tend to use continuous representations (with some exceptions; e.g., Flow-Omni (Yuan et al., 2024) is a speech+text LM that generates continuous speech representations, and DiscreteSLU (Shon et al., 2024) is a speech-aware text LM that uses discrete speech token input). Finally, temporal compression (e.g., deduplication or BPE) is optionally applied to reduce the sequence length.

### 3.1.1 Continuous Features

To extract informative representations from raw waveforms, a speech representation model—either a learned encoder or a digital signal processing (DSP) feature extractor—converts speech into continuous features. These continuous features may include:

1. Traditional spectrogram features, such as mel filter bank features (Huang et al., 2001).

2. Hidden representations from self-supervised learning-based (SSL) speech encoders, such as wav2vec 2.0 (Baevski et al., 2020), HuBERT (Hsu et al., 2021), WavLM (Chen et al., 2022), or XEUS (Chen et al., 2024b).

3. Hidden representations from supervised pre-trained models, such as Whisper (Radford et al., 2023) or USM (Zhang et al., 2023b).

4. Hidden representations from neural audio codec models, such as SoundStream (Zeghidour et al., 2022) or EnCodec (Défossez et al., 2023).

### 3.1.2 Discrete Tokens

We divide discrete speech tokens into two main categories, "phonetic tokens" and "audio codec tokens", based on how they are learned and their perceived properties.[4]

**Phonetic tokens**   Phonetic tokens are most commonly derived by quantizing self-supervised speech representations (Mohamed et al., 2022) or, occasionally, supervised encoders from a pre-trained ASR model (as in CosyVoice (Du et al., 2024), GLM-4-Voice (Zeng et al., 2024), and WhisperSpeech (WhisperSpeech, 2024)). These quantized representations tend to have strong similarity to phonetic units (Sicherman & Adi, 2023). The pioneering pure speech LM GSLM (Lakhotia et al., 2021) treated these tokens as "pseudo text," allowing the model to function similarly to text-based LMs and to generate phonetically coherent speech. These tokens, learned directly from raw speech, formed a foundation for subsequent research, often described as "textless NLP" (Polyak et al., 2021; Lakhotia et al., 2021; Kharitonov et al., 2022b; Nguyen et al., 2023; Hassid et al., 2023).

The process of deriving phonetic tokens involves several design decisions, such as selecting a pre-trained representation model, which layer's representations to quantize, and the number of clusters. Different layers in SSL speech models tend to encode different types of information (Hsu et al., 2021; Mousavi et al., 2024b; Pasad et al., 2023). Representations from layers rich in phonetic information are typically chosen for extracting phonetic tokens. The number of clusters is also a key hyperparameter, and is often optimized based on downstream task performance or a proxy task (e.g., the ABX task (Schatz et al., 2013)). Too few clusters may result in a loss of fine-grained phonetic information, while too many clusters may risk encoding undesirable speaker-specific features (Lakhotia et al., 2021; Kharitonov et al., 2022a).

**Audio codec tokens**   In contrast to phonetic tokens, audio codec tokens are intended to capture more detailed acoustic characteristics. These tokens are derived from neural codec models, which were originally designed for audio compression and therefore for faithful reconstruction of audio from its encoding (Zeghidour

---

[4]In some literature, phonetic tokens are referred to as "semantic tokens" (Borsos et al., 2023a). However, it's important to clarify that "semantic" in this context does not imply the traditional linguistic meaning, as these tokens are more akin to phonetic units (Sicherman & Adi, 2023) and do not typically carry semantic content. Therefore, we use the term "phonetic tokens" throughout this paper.

et al., 2022; Défossez et al., 2023). Although audio codecs were initially intended for compression, their intermediate discrete representations have proven valuable as tokens in SLMs (Borsos et al., 2023a).

Neural codecs comprise three components: an encoder that converts raw audio into frame-level features, a vector quantization (VQ) module that converts these features into discrete tokens, and a decoder that reconstructs audio from the codec tokens. Typically, the main encoder-decoder backbone is a convolution-dominant architecture (e.g., Encodec (Défossez et al., 2023)), but some recent work has introduced transformers as intermediate layers (e.g., Mimi (Défossez et al., 2024)) or transformer-only architectures to reduce computation (e.g., TS3-Codec (Wu et al., 2025b)).

For the VQ module, a commonly used approach is residual vector quantization (RVQ) (Gray, 1984; Zeghidour et al., 2022), which generates several discrete tokens for each time step, corresponding to multiple (hierarchical) levels of granularity (with the first level encoding most of the information, and the remaining levels encoding residual information). Decoding such multi-codebook speech tokens typically requires additional design considerations for the SLM architecture and decoding strategy (Borsos et al., 2023a; Chen et al., 2025b; Yang et al., 2024a). Alternatively, several single-codebook quantization codecs (Wu et al., 2025b; Xin et al., 2024) have been developed to simplify decoding in SLMs. Section 3.3 describes decoding strategies in detail.

**Comparison of token types** Early SLMs, such as GSLM, typically used phonetic tokens, which reduce speaker-specific information (Polyak et al., 2021). This reduction allows language models to focus primarily on spoken content, making them more effective for understanding tasks like detecting words vs. nonwords or syntactic correctness (Lakhotia et al., 2021; Hassid et al., 2023) and for applications such as speech-to-speech translation (Lee et al., 2022a;b). In contrast, audio codec tokens are frequently used in tasks where preserving speaker identity and acoustic details is crucial (Chen et al., 2025b).

Recently, several tokenization approaches have explored hybrid strategies that combine phonetic and acoustic tokenization. Some examples include distilling pre-trained SSL representations into RVQ codebooks (Zhang et al., 2024b; Défossez et al., 2024), integrating phonetic or textual information during the tokenization process (Tseng et al., 2025; Har-Tuv et al., 2025), and incorporating LM-based objectives into tokenization learning (Turetzky & Adi, 2024).

In SLMs, two important factors in the choice of tokenization are the token bit rate, which impacts efficiency, and the token quality, which is related to generation quality and suitability for downstream tasks. Several benchmarks have been established to evaluate different types of tokens (Shi et al., 2024; Wu et al., 2024c; Mousavi et al., 2024a; Wu et al., 2024b). Codec-SUPERB (Wu et al., 2024c), the first neural audio codec benchmark, evaluates the quality of resynthesized audio, using subjective metrics and pre-trained downstream models for comparison. DASB (Mousavi et al., 2024a) evaluates tokenization methods by using the extracted tokens for various downstream tasks. ESPnet-Codec (Shi et al., 2024) is an open-source framework that functions as both a toolkit for neural codec training and a platform for evaluation in a controlled setting.

Designing efficient and effective tokens for SLMs remains an active area of research; for more detailed surveys, see (Guo et al., 2025; Mousavi et al., 2025).

### 3.1.3 Temporal Compression

Quantized speech feature sequences are often very long due to their high frame rates. To mitigate the challenges this poses for language modeling, several techniques are typically applied within the encoder to reduce the sequence length. One such technique is "deduplication", which merges consecutive identical tokens into a single token (as shown in Figure 3). However, this approach loses information about the duration of individual tokens. To address this issue, some approaches use multi-stream modeling to capture token duration in a separate stream (Nguyen et al., 2023), while others introduce duration prediction when generating output speech (Kreuk et al., 2022; Maimon & Adi, 2023; Lee et al., 2022a). Additionally, byte pair encoding (BPE) (Gage, 1994) is also sometimes applied to the discrete token sequences to capture recurring patterns (Wu et al., 2023a; Shen et al., 2024).

### 3.2 Speech Modality Adapter

In many SLMs (especially speech-aware text LMs), the speech encoder (Section 3.1) and the sequence model (Section 3.3) are initially developed separately and then combined. It is therefore necessary to somehow align the output of the speech encoder with the expectations of the sequence model, and this is the role of the modality adapter. The modality adapter is typically trained on downstream tasks or, in the case of speech+text LMs, as part of pre-training (see Section 4 for more details on training).

If the output sequence of the speech encoder is not very long, the modality adapter can be as simple as a linear layer, which transforms the encoder output into the embedding space of the sequence model. This typically occurs when the encoder produces discrete tokens with temporal compression. On the other hand, if the output of the encoder is too long, the adapter is typically also responsible for shortening the sequence. This usually happens when the encoder produces continuous representations without discretization or temporal compression. Shortening the input sequence simplifies both training and inference for the sequence model. Furthermore, since the sequence model often processes both text and speech inputs, it is helpful for the token rate of the speech sequence to roughly match that of the text sequence.

Common adapters include:

**Linear transformation / vocabulary expansion** A straightforward way to integrate speech into a pre-trained sequence model is by applying a linear transformation to the speech representation $H^{\text{sp}}$. In this approach, the adapter $\text{Adp}^{\text{sp}}()$ is defined as a linear transformation. This method is commonly used when the speech encoder produces discrete token outputs. This approach can be interpreted as vocabulary expansion, where the speech tokens are treated as additional tokens in the sequence model's vocabulary. Their embeddings are then learned during subsequent task-oriented training. For example, in SpeechGPT (Zhang et al., 2023a), the vocabulary of the sequence model LLaMA (Touvron et al., 2023a) is expanded to include HuBERT-based phonetic tokens. Similarly, in Mini-Omni2 (Xie & Wu, 2024b), the sequence model Qwen (Bai et al., 2023) incorporates eight layers of acoustic codec tokens alongside standard text tokens.

**CNN with strides**: Convolution layers with strides reduce the sequence length while preserving temporal information (Lu et al., 2024), which is essential for tasks that require such information, like ASR (Wu et al., 2023b). A special case of this adapter type is pooling layers with strides (Chu et al., 2023).

**Connectionist temporal classification (CTC)-based compression**: This method compresses $H^{\text{sp}}$ (Eq. 3) according to the posterior distribution from a CTC-based speech recognizer (Gaido et al., 2021). CTC (Graves et al., 2006), a commonly used approach for ASR, assigns each time step a probability distribution over a set of label tokens, including a blank ("none of the above") symbol. The time steps with high non-blank probabilities indicate segments that are likely to carry important linguistic information. CTC compression aggregates the frame-level labels, specifically by merging repeated non-blank labels and removing blanks. This approach produces a compressed representation intended to retain the relevant content of the original sequence while significantly reducing its length (Wu et al., 2023b; Tsunoo et al., 2024).

**Q-Former**: The Q-Former (Li et al., 2023) is an adapter that produces a fixed-length representation by encoding a representation sequence of arbitrary length into $M$ embedding vectors, where $M$ is a hyperparameter (Lu et al., 2025a).

Let the input speech representation sequence be:

$$H^{\text{sp}} = \{h_1, h_2, \ldots, h_{L'}\}, \quad h_l \in \mathbb{R}^{d'}, \tag{10}$$

where $L'$ is the sequence length and $d'$ is the dimension of the embeddings.

To achieve a fixed-length representation, Q-Former introduces $M$ trainable query embeddings:

$$Q = \{q_1, q_2, \ldots, q_M\}, \quad q_m \in \mathbb{R}^{d'}. \tag{11}$$

These queries interact with $H^{SP}$ via a cross-attention mechanism:

$$\text{Attn}(Q, H^{\text{sp}}) = \text{softmax}\left(\frac{QW_Q(H^{\text{sp}}W_K)^T}{\sqrt{d'}}\right) H^{\text{sp}}W_V, \tag{12}$$

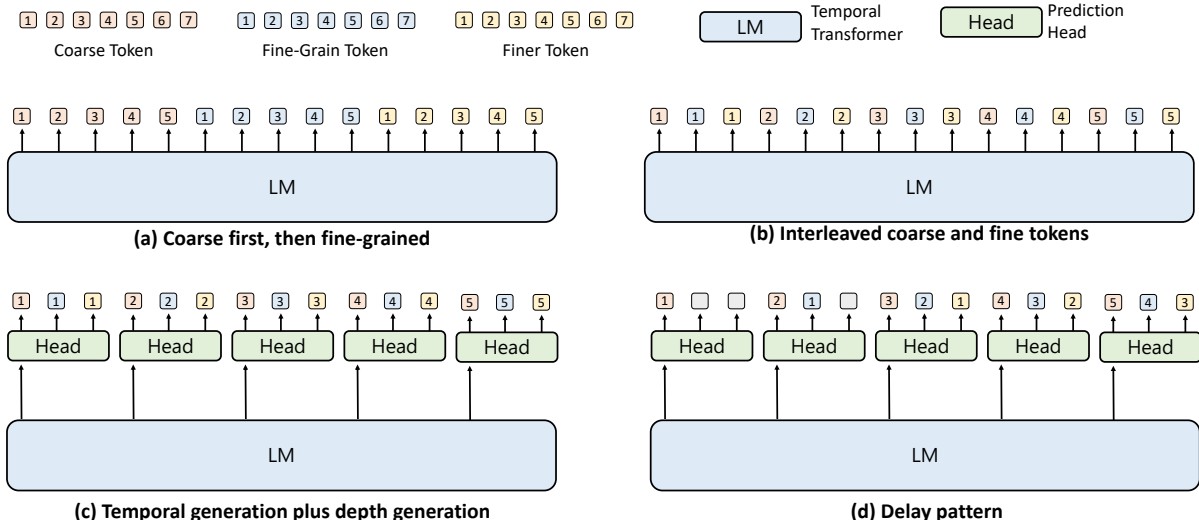

Figure 4: Hierarchical generation strategies (see Section 3.3.1).

where $W_Q$, $W_K$, and $W_V$ are learnable projection matrices. The result is a sequence of $M$ embeddings.

In some approaches, instead of directly encoding the entire utterance into $M$ vectors, a *window-level Q-Former* is applied (Yu et al., 2024; Pan et al., 2024; Tang et al., 2024) to retain temporal information. In the window-level Q-Former, the input embedding sequence is segmented, and the Q-Former is applied to each segment.

Lu et al. (2024) compare the Q-Former with CNN-based modality adapters in a speech-aware text LM, finding that the Q-Former produces better performance on the Dynamic-SUPERB benchmark (Huang et al., 2024) (see Section 7 for more on this and other SLM benchmarks).

**AlignFormer**: AlignFormer (Fan et al., 2025) combines a CTC compressor (Gaido et al., 2021) with a Q-Former (Li et al., 2023). When the LLM backbone is frozen, Fan et al. (2025) find that AlignFormer enables zero-shot instruction-following capabilities for speech QA tasks using only ASR training data. Additionally, AlignFormer surpasses Q-Former in instruction-following performance across multiple datasets and tasks, including SQA and speech translation.

**Others**: Several other methods have been developed for the modality adapter. For example, LTU-AS uses time and layer-wise transformers (TLTR) (Gong et al., 2023a), WavLLM (Hu et al., 2024) uses a bottleneck adapter (Houlsby et al., 2019), and other approaches use multi-layer perceptron (MLP) adapters (Fang et al., 2025a; Microsoft et al., 2025).

## 3.3 Sequence Model

In principle, generating speech tokens is similar to generating text tokens. However, there are some important differences. Unlike text tokens, audio tokens may include a mix of coarse and fine tokens — for example, phonetic tokens as coarse tokens and audio codec tokens as fine tokens. For TTS models, using phonetic tokens as an intermediate representation rather than directly mapping text to audio codec tokens provides better supervision and reduces modelling complexity. For example, SPEAR-TTS (Kharitonov et al., 2023), which uses phonetic tokens, has similar performance to VALL-E (Chen et al., 2025b) while requiring significantly less parallel text-to-speech training data. The same concept applies to SLMs, as described in Section 3.3.1 below. In addition to generating multiple speech token types, some SLMs combine text and speech token generation to enhance the linguistic quality of the generated speech, as described in Section 3.3.2.

### 3.3.1 Hierarchical Token Generation

There are several decoding strategies for hierarchical modeling of multiple types of tokens of different granularity (see the discussion in Section 3.1.2), shown in Figure 4. Phonetic tokens and first-layer codec tokens (e.g., Mimi (Défossez et al., 2024) and SpeechTokenizer (Zhang et al., 2024b)) can be regarded as coarse tokens in Figure 4, while the remaining layers of codec tokens can be regarded as "fine-grained" and "finer" tokens.[5] We categorize common decoding strategies into the following types:

**Coarse first, then fine-grained (Figure 4 (a)):**  In this approach, coarse tokens for all time steps are generated first, followed by fine-grained tokens conditioned on the coarse tokens, and finally the finer tokens. For example, AudioLM (Borsos et al., 2023a) uses such a three-stage autoregressive process (which can be modeled by different LMs), first predicting phonetic tokens (the coarse tokens in Figure 4 (a)), then first-layer audio codec tokens (the fine-grained tokens in Figure 4 (a)) conditioned on the phonetic tokens, and finally the remaining-layer codec tokens (the finer audio codec tokens in Figure 4 (a)). SoundStorm (Borsos et al., 2023b) replaces the second and third stages of AudioLM with a non-autoregressive prediction approach inspired by MaskGIT (Chang et al., 2022), which begins with masked tokens and then non-autoregressively predicts subsets of tokens in rounds based on confidence scores. Similarly, VALL-E (Chen et al., 2025b) uses an autoregressive model to predict the first-layer audio codec tokens based on the corresponding phoneme transcriptions, and then uses a non-autoregressive model to generate the remaining codec token sequences.

**Interleaved coarse and fine tokens (Figure 4 (b)):**  In this decoding strategy, the three types of tokens——coarse, fine-grained, and finer——are aligned along the time axis and interleaved for generation. When predicting tokens for time step $t$, the coarse, fine-grained, and finer tokens corresponding to $t$ are generated sequentially. SpiRit-LM (Nguyen et al., 2025) applies this interleaved structure (interleaved phonetic, pitch and style tokens) to enhance speech expressiveness, and it also includes text tokens in addition to the multiple types of speech tokens.

**Temporal generation plus depth generation (Figure 4 (c)):**  This strategy employs a large transformer LM (indicated in blue) to model inter-frame information along the time axis and a small transformer head (indicated in green) to capture intra-frame correlations. The small transformer head predicts multi-layer tokens——where fine-grained and finer tokens correspond to different layers of audio codec tokens——within the same time step. For example, UniAudio (Yang et al., 2024a) adopts this approach, inspired by RQ-Transformer (Lee et al., 2022c) and the MegaByte transformer model (Yu et al., 2023). Moshi (Défossez et al., 2024) also adopts a similar strategy.

**Delay pattern (Figure 4 (d)):**  In this approach, there are time delays (of 1 or more time steps, depending on the design) between coarse and fine tokens. Such time delays allow the model to utilize look-ahead, that is to use future coarse tokens as conditions when generating fine tokens. Figure 4 (d) shows an example where the delay is 1. The large blue LM in Figure 4 (d) autoregressively outputs the temporal embeddings and feeds them into the green prediction head, which then predicts the coarse token and the delayed fine token in parallel. For example, pGSLM (Kharitonov et al., 2022b) introduces a delay between the phonetic tokens and the "expressive" (pitch and duration) tokens.

Using multiple stages of prediction can achieve both high audio quality and long-term consistency (Borsos et al., 2023a). However, this approach makes the decoding strategy complex and increases latency, making it less suitable for real-time applications like spoken dialogue generation. To address this drawback, there is a growing literature on tokenization methods that reduce the number of tokenization layers. One example is to combine the generation of phonetic and audio codec tokens into a single tokenizer by distilling the information from phonetic tokens directly into the audio codec (Défossez et al., 2024; Zhang et al., 2024b).

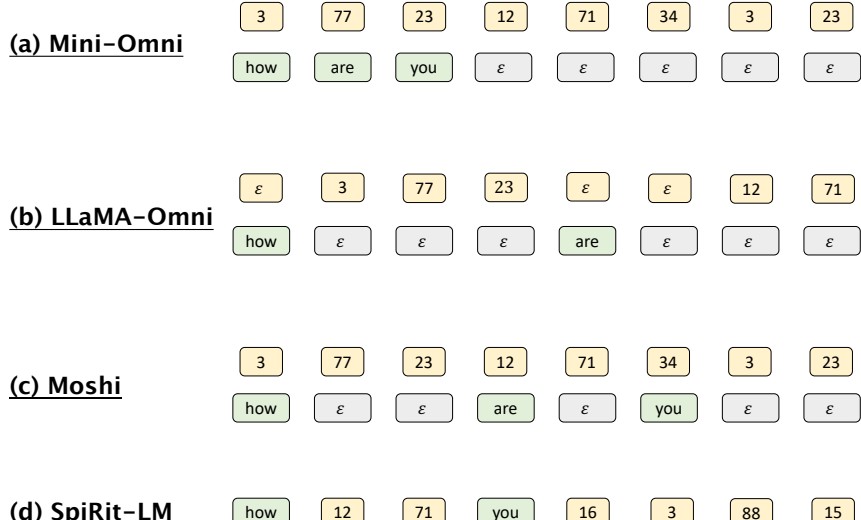

Figure 5: Text and speech hybrid generation, exemplified by four representative models (a)-(d) (see Section 3.3.2). Green and yellow boxes represent text and audio tokens, respectively.

### 3.3.2 Text and speech hybrid generation

Some sequence generation models use text tokens as the first-layer tokens, followed by speech tokens (either phonetic or audio codec tokens). This approach allows the SLM to leverage the knowledge from a pretrained text LM, which is typically trained on much larger-scale data than the available speech data, to improve the factuality and linguistic quality of generated speech. As shown by Défossez et al. (2024), this approach can improve language modeling quality, as measured by negative log-likelihood scores measured by an external text LM.[6] Another advantage to this approach is that during the process of developing the SLM, it is straightforward to separately evaluate both the correctness of the output content and the speech generation quality.

Text and speech tokens operate on different scales: Speech has a fixed sampling rate (unless tokenized into tokens of different durations), whereas text tokens do not. A text token sequence is also usually shorter than the corresponding speech token sequence. Hybrid decoding needs to address the issue of differing lengths and, ideally, also to achieve temporal alignment (synchronization). There are four main types of text-speech hybrid generation used in recent SLMs, shown in Figure 5. We use four representative models to illustrate the ideas.

The first type (Figure 5 (a)) addresses the mismatch in sequence lengths by adding padding tokens after the text token sequence. In this setup, the text sequence ends first, and the generated text then guides the generation of the speech token sequence, making the process similar to text-to-speech (TTS). A representative example is Mini-Omni (Xie & Wu, 2024a), which adopts a delayed decoding approach (Copet et al., 2023).

The second type (Figure 5 (b)) involves adding fixed padding tokens after each text token to extend the text token sequence. Padding tokens are also added between speech tokens so that both sequences have the same length. An example of this approach is LLaMA-Omni (Fang et al., 2025a): After predicting the text tokens (padded with fixed-length padding tokens), it then predicts the phonetic token sequence based on the padded text (where the phonetic token sequence is shorter than the padded text token sequence), and finally the CTC loss is applied to the phonetic token sequence to guide training.

The third type (Figure 5 (c)) dynamically adjusts the number of padding tokens between text tokens. The generative model learns to insert these dynamic padding tokens to make the text and speech sequences

---

[5]In Figure 4 and in running examples, we assume 3 granularity levels. In practice, a sequence model can include a smaller or larger number of levels, and this number is also an important design choice.

[6]In this case, LiteLlama-460M-1T, https://huggingface.co/ahxt/LiteLlama-460M-1T

the same length. During training, time-aligned text-speech paired data is used to construct padded text sequences that match the lengths of the speech sequences. Moshi (Défossez et al., 2024) is an example of this model type.

The fourth type (Figure 5 (d)) interleaves speech and text tokens in a single sequence, with text-speech token alignments derived in advance. SpiRit-LM (Nguyen et al., 2025) is an example of this approach, using time-aligned text-speech paired data for training. GLM-4-Voice (Zeng et al., 2024; 2025) uses a pre-trained text-to-token model to generate audio tokens from text, and interleaves the generated audio with text tokens.

### 3.4 Speech Decoder

The speech decoder converts speech representations——whether continuous features, phonetic tokens, or audio codec tokens——back into waveforms. The speech decoder can take various forms:

1. Vocoder (Kong et al., 2020) for continuous features, similar to those used in traditional synthesis systems. For instance, in Spectron (Nachmani et al., 2024), a generated Mel spectrogram is synthesized into audio using the WavFit vocoder (Koizumi et al., 2023).

2. Unit-based vocoder (Polyak et al., 2021) based on HiFi-GAN (Kong et al., 2020) for phonetic tokens. These vocoders take phonetic tokens as inputs and optionally combine them with additional information to improve synthesis quality. For example, when phonetic tokens are deduplicated, a duration modeling module is often included in the vocoder (Lakhotia et al., 2021).

3. Codec decoder (Guo et al., 2025). When the SLM generates audio codec tokens, these tokens can be input directly into the corresponding pre-trained codec decoder (without additional training) to generate the waveform.

## 4 Training Strategies

We divide the training phases of SLMs into *pre-training* and *post-training*, analogously (but not identically) to the typical division in text LLM training. For text LMs, pre-training refers to training a model of $p(text)$ with a next-token prediction objective, whereas post-training is task-oriented and directly facilitates alignment and improved performance on tasks. Since many SLMs start with a post-trained text LLM, we consider this to be a part of SLM pre-training. More precisely, we define *pre-training* and *post-training* of SLMs as follows:

**Pre-training:** We define pre-training as any training strategy that *does not have the explicit goal of enabling the model to perform a wide variety of downstream speech tasks*. In particular, pre-training does not directly aim to make a model a universal speech processing system (but, as described below, it can include training for specific tasks). In the context of SLMs, which handle multiple modalities, pre-training often involves a multi-stage process. This may include initial next-token-prediction training on text, followed by continual training on speech continuation tasks using large unlabeled speech corpora. Continual training may enhance the capabilities of SLMs for specific tasks, but the resulting pre-trained SLMs remain limited in scope and are still far from being universal models. Text LLM post-training before adding the speech modality is also considered pre-training of the SLM, since it does not target universal *speech* processing.

**Post-training:** Post-training refers to training strategies *focused on enabling a variety of downstream tasks, with the goal of producing a more or less universal speech processing system*. The tasks can be either predefined with specific task specifiers or dynamically defined through text or spoken language prompts. Instruction tuning (Tang et al., 2024; Pan et al., 2024; Shu et al., 2023) and preference optimization (Lin et al., 2025c) are examples of post-training.

### 4.1 Pre-training strategies

Here we focus on speech-specific pre-training strategies; that is, we do not discuss the training of any text LM that is included in SLM pre-training.

### 4.1.1 Generative pre-training

**Pure speech pre-training** $p(speech)$ Pure speech pre-training refers to training a model of $p(speech)$ from unlabeled speech (a pure speech LM in Table 1), typically by tokenizing the speech and modeling the token sequences with an autoregressive model trained to perform *next speech token prediction*. The success of SSL speech representation models facilitated the development of pure speech LMs, because SSL models provided high-quality representations that could easily be quantized to produce tokenized speech. Pure speech LMs, such as GSLM (Lakhotia et al., 2021), typically use discrete phonetic speech tokens as the basic language modeling units, but *non-phonetic information*—such as duration and pitch—can also be incorporated (Kharitonov et al., 2022b; Nguyen et al., 2023) to enhance prosody modeling. Section 5.1 expands on pure speech LMs.

**Joint speech and text pre-training** $p(text, speech)$ This approach refers to pre-training SLMs on aligned speech and text to model $p(text, speech)$ (speech+text LM in Table 1) (Chou et al., 2023; Nguyen et al., 2025; Défossez et al., 2024; Tian et al., 2025a). The speech and text sequences may be interleaved (Chou et al., 2023; Nguyen et al., 2025) or treated as two channels in a dual-channel approach (Défossez et al., 2024), as detailed in Section 3.3.2.

**Continual pre-training following text pre-training** $p(text)$ Continual pre-training (Ke et al., 2023) refers to the process of further training a pre-trained model on additional data that is domain- or modality-specific, but still not task-oriented. This intermediate step allows the model to adapt to new domains or datasets while (hopefully) retaining its general-purpose capabilities (Tang et al., 2024; Chu et al., 2023; Hassid et al., 2023). A common example of continual pre-training for SLMs is to start with a text LLM pre-trained with a next token prediction objective (a pure text LM in Table 1) and then further pre-train it with speech data to improve its performance on the speech continuation task (Hassid et al., 2023).

### 4.1.2 Conditional pre-training $p(text|speech)$

While generative pre-training is the most common form of pre-training, it is also in principle possible to initialize SLM training from a *conditional* model. For example, the pre-trained conditional model could be an encoder-decoder speech recognizer (Chan et al., 2016) or a joint speech recognition and translation model like Whisper (Radford et al., 2023) or OWSM (Peng et al., 2023b; 2024b; 2025a) that uses task specifier tokens to indicate the desired task. Such models can be trained on massive amounts of transcribed speech, and therefore have already learned to align the speech and text modalities. Although these models are trained for very specific tasks, they have shown promise as an initialization for instruction-following models that perform a wider range of understanding tasks (Lai et al., 2023; Arora et al., 2024).

### 4.1.3 Aligning speech and text modalities

For SLMs that combine pre-trained text LMs and speech encoders, another pre-training strategy is to align the speech and text modalities in a task-independent way.

**Implicit alignment** Speech and text modalities can be implicitly aligned through techniques such as the "modal-invariance trick" (Fathullah et al., 2024) or behavior alignment (Wang et al., 2023a). The idea is that the model should produce identical responses regardless of the input modality, provided the input conveys the same meaning. This approach often utilizes ASR datasets. The text transcript is input to a text LLM to generate a text response, while the corresponding speech recording is input into the SLM, which is trained to generate the same text response. Another useful idea for implicit alignment is training spoken LLMs for audio captioning, where a spoken LLM takes audio as input and outputs its description. It has been observed that training a spoken LLM solely through audio captioning can generalize to tasks it has never seen during training (Lu et al., 2024; 2025a).

**Explicit alignment** Speech and text modalities can also be explicitly aligned by matching speech features to corresponding text embeddings, via optimization of appropriate distance/similarity measures. For example, Wav2Prompt (Deng et al., 2025) and DiVA (Held et al., 2025) align modalities by minimizing the $L_2$

Table 2: Examples of instructions for speech-related tasks used in SLM instruction tuning.

| Task | Examples of Instructions |
|---|---|
| Speech recognition | Recognize the speech and give me the transcription. (Tang et al., 2024) 
 Repeat after me in English. (Grattafiori et al., 2024) |
| Speech translation | Translate the following sentence into English. (Grattafiori et al., 2024) 
 Recognize the speech, and translate it into English (Chu et al., 2023) |
| Speaker recognition | How many speakers did you hear in this audio? Who are they? (Tang et al., 2024) |
| Emotion recognition | Describe the emotion of the speaker. (Tang et al., 2024) 
 Can you identify the emotion? Categorise into: sad, angry, neutral, happy (Das et al., 2024) |
| Question answering | What happened to this person? (Wang et al., 2023b) 
 Generate a factual answer to preceding question (Das et al., 2024) 
 What medicine is mentioned? Briefly introduce that medicine. (Peng et al., 2025b) |

distance between speech features and the token embeddings of their transcripts in a text LLM while keeping the text embeddings fixed.

## 4.2 Post-training strategies

The pre-training of LLMs and SLMs in Section 4.1 enables the modeling of the general data distributions of text $p(text)$ or speech $p(speech)$, the joint distribution of speech and text $p(text, speech)$, or a conditional distribution $p(text|speech)$ based on specific pre-training tasks. However, pre-trained models still often lack the capability to solve a large range of downstream spoken language tasks, to follow natural language instructions, or both. In the post-training phase, carefully curated datasets are used to train SLMs to generate desired outputs or perform tasks, typically specified using natural language instructions.

### 4.2.1 Task-specific training

While the eventual goal is to handle arbitrary tasks within a single model via instructions, some SLM approaches begin with a simpler post-training setting: multi-task training with task specifiers $p(\cdot|speech, \langle task\ specifier\rangle)$. In this approach, the pre-trained SLM is fine-tuned for a predefined set of target tasks. Qwen-Audio is an example of such an approach (Chu et al., 2023). Task-specific training can then be followed by instruction tuning or other post-training approaches, described below.

### 4.2.2 Instruction tuning $p(\cdot|speech, instruction)$

Instruction tuning of SLMs has been inspired by, and closely follows, successful approaches for text LLM instruction tuning. Instruction-tuning data typically consists of a speech recording, an instruction that describes the speech task, and the ground-truth output. During instruction tuning, the instructions are appended to the speech recording as inputs to the SLM, which is trained to generate the corresponding ground-truth output. Depending on the model design, the instruction can be in either text format (Tang et al., 2024; Pan et al., 2024) or speech format (Shu et al., 2023) . In both cases, it has been found that SLMs trained on diverse instruction-tuning data can perform tasks unseen during the instruction-tuning phase (Tang et al., 2024; Das et al., 2024; Peng et al., 2025b). Table 2 shows examples of instructions for various speech tasks taken from existing instruction tuning sets.

Instruction-tuning data can be generated through various methods:

**Conversion of task-specific data to instructions** Task-specific data (Section 4.2.1) can be adapted to the instruction-tuning format by replacing task-specific tags with natural language instructions (Tang et al., 2024). LLMs can be used to rephrase those instructions to increase diversity (Arora et al., 2024).

**Speech-based question answering (SQA) data** Such data is typically generated using LLMs such as ChatGPT. In this process, the transcript of a speech recording is provided as input to an LLM, which is

instructed to generate a question-answer pair related to the speech content in text format (Tang et al., 2024; Gong et al., 2024; Peng et al., 2025b). To incorporate additional context, supplementary textual descriptions about attributes such as the speaker, gender, age, emotion, and noise level may also be provided to the LLM (Yang et al., 2024b). During training, the speech recording and the corresponding LLM-generated question are provided as inputs, and the model learns to predict the LLM-generated answer.

**Synthesis of text instruction-tuning data**   Speech instruction-tuning data can also be created by applying TTS to existing textual instruction-tuning or user-assistant conversation datasets (Peng et al., 2025b). In this approach, either a subset or the entirety of the user's input is converted to speech. This type of data encompasses a wide variety of instruction types and response styles. The answers tend to be more descriptive than the ones in SQA and are presented in diverse textual formats, such as markdown.

**Compositional instructions**   Instructions for individual tasks can be combined to form more complex instructions, which improves the performance on more challenging tasks. For example, an SLM can be instructed to first perform speech recognition and then speech translation conditioned on the ASR hypothesis (Hu et al., 2025).

### 4.2.3 Chat SLM training

In addition to the general post-training methods mentioned above, an important specific application setting that is gaining attention is conversational or chat SLMs, which require carefully curated training data and tailored training strategies. The development of chat SLMs has involved two main directions: (1) building a speech-aware text LM based on a text LM with chat capabilities, and (2) creating a pure speech LM or a speech+text LM that can handle speech-input-to-speech-output conversations. For (1), one approach is to use a more powerful text LLM to generate text-based conversations centered around a speech recording and use this pseudo-conversation data to fine-tune the SLM (Chu et al., 2023). For (2), a common approach is to use a TTS system to generate speech conversation data from text datasets (Zhang et al., 2023a; Défossez et al., 2024). If available, real speech conversation data can be used to provide more natural, spontaneous behaviors——such as pauses and interruptions——that occur in real conversations, though such data is often noisier and requires careful preprocessing (Défossez et al., 2024). Recent efforts have included post-training conversational SLMs with tool-usage capabilities to enhance response accuracy (Arora et al., 2025a). Section 6 addresses conversational SLMs in greater detail.

### 4.3 Other training details

In addition to the basic strategies discussed in Sections 4.1 and 4.2, several additional training methods have proven useful for building universal SLMs, including:

**Parameter-efficient fine-tuning (PEFT)**   Since pre-training equips LMs with strong text and/or speech generation capabilities, it is possible to not update the entire LM during the fine-tuning phase. Common strategies include: (1) freezing both the sequence model $Seq$ and the speech encoder $\text{Enc}^{\text{sp}}$ (see Figure 2) and updating only the parameters of the adaptor $\text{Adp}^{\text{sp}}$ (Wang et al., 2023b), and (2) adding to the sequence model a set of parameter-efficient trainable adapter modules, which are trained alongside $\text{Adp}^{\text{sp}}$ (Tang et al., 2024).

**Progressive fine-tuning**   A key objective in training speech-aware LMs and speech+text LMs is to align the hidden representations of speech and text, in order to leverage the generation and reasoning capabilities of the pre-trained LLM. To achieve this goal, it is common to kick off post-training with content-heavy tasks, such as ASR, and gradually progress to tasks that require a more comprehensive understanding of additional information embedded in speech (e.g. emotion recognition) (Tang et al., 2024) or tasks composed of multiple sub-tasks (Hu et al., 2024). To stabilize training, a small number of components is updated during the initial training stage, while more are updated later.

A similar strategy has been adopted in work on conversational SLMs (Défossez et al., 2024). The intrinsic properties of human conversations present several challenges for building such SLMs. For instance, speech

from different speakers often overlaps, rather than following a strictly turn-based structure. Therefore, training on real human conversations is essential. However, collecting spoken human-human conversation datasets is difficult, and publicly available datasets are often limited in size. To address these challenges, these models may be first trained on multichannel audio data before being fine-tuned on real human conversation data. See Section 6 for more details about such models.

**Experience replay of pre-training data during post-training** To prevent the catastrophic forgetting phenomenon (Kirkpatrick et al., 2017), reusing pre-training data during the post-training stage can help the SLM retain important capabilities learned during pre-training, such as the reasoning abilities of LLMs associated with text instruction-tuning (Peng et al., 2025b). Another approach to prevent forgetting is to use data generated by the original text LLM during post-training (Lu et al., 2025a).

**Preference optimization** Reinforcement learning from human feedback (RLHF) (Ouyang et al., 2022) has become a key method for aligning LLM generations with human preferences. This method was used in Qwen2-Audio (Chu et al., 2024) to improve the quality of responses generated by the speech-aware text LM. On the other hand, Align-SLM (Lin et al., 2025c) was the first to apply this method to a pure SLM and used AI-generated feedback as a substitute for human evaluation to make the process more cost-efficient, and this strategy has also proven useful in speech+text LMs as well Wu et al. (2025a). Recently, Maimon et al. (2025a) analyzed the performance of pure SLM fine-tuning using direct prefenrece optimization as a function of fine-tuning examples.

## 5 Survey of representative SLMs

The previous sections have described the components, design choices, and training strategies for SLMs. In this section, we review how these choices have been instantiated in existing SLMs in the literature, categorized according to the classes in Table 1. For each category, we provide a brief review of the model design and highlight example models in the literature. Note that, aside from the examples listed, there are many SLMs with similar configurations in each category which we are unable to introduce in detail; Tables 4 and 5 provide a more extensive catalogue of models and references, and Figure 1 provides a timeline to place them in historical context.

We also note that some of the multi-modal LLMs developed by industry groups, such as GPT-4o (OpenAI, 2024) and Gemini (Gemini Team et al., 2024), can also be seen as generalizations of speech+text LMs or speech-aware text LMs. These models have conversation capabilities and include text, speech, general audio, and visual inputs and outputs. However, details of their model configurations and training strategies have not been publicly released, and rigorous evaluations of these models have focused on limited speech tasks such as ASR and ST. For these reasons, it is important that future work complements these high-impact industry models with open-source, reproducible approaches.

### 5.1 Pure Speech LM

Pure speech LMs are trained on unlabelled speech data to model $p(speech)$, and are the spoken analogue of pre-trained generative text LMs like GPT (Radford et al., 2019; Brown et al., 2020). In order to apply the next-token-prediction objectives of LM training to continuous speech data, pure speech LMs typically use discrete speech representations (Section 3.1) as the units for sequence modeling. One of the earliest approaches in this category is the generative spoken language modeling (GSLM) work of Lakhotia et al. (2021). GSLM relies on a pre-trained HuBERT model $Enc^{sp}()$ to convert speech signals into phonetic tokens $H^{sp}$ and uses an embedding layer $Adp^{sp}()$ to map token IDs to continuous vectors. The model is trained to autoregressively predict the ID of the next speech token, and the predicted tokens are converted back to audio by a unit-based vocoder $Dec^{sp}()$. GSLM's demo[7] shows anecdotally that this initial model can generate speech continuations with locally acceptable syntax given a speech prompt of a few seconds, but does not in general produce coherent long-form speech.

---

[7]https://speechbot.github.io/gslm

Since GSLM initiated this line of research, follow-up work has improved on it in several ways. For example, the prosody-aware generative spoken language model (pGSLM) (Kharitonov et al., 2022b) predicts prosodic information like pitch and duration jointly with phonetic tokens for better expressiveness. TWIST (Hassid et al., 2023) is similar to GSLM but the model is initialized with a pre-trained text LLM (OPT (Zhang et al., 2022)), which improves lexical and syntactic language modeling performance, as measured by the sWUGGY and sBLIMP metrics (see Section 7.1 for more details of the evaluation). Another approach (Algayres et al., 2023) uses *longer* lexical units instead of phonetic tokens, significantly reducing memory consumption. Audio-LM (Borsos et al., 2023a) introduced hierarchical generation of multiple types of tokens for pure speech LMs, using a coarse-to-fine strategy that generates phonetic tokens and audio codec tokens in order (Section 3.3.1). SpeechSSM (Park et al., 2025) uses a state-space model (Gu et al., 2022) rather than a Transformer, producing more natural and semantically consistent long-form (minutes-long) speech generation. Flow-SLM (Chou et al., 2025) and CALM (Simon et al., 2025) generate continuous representations rather than or in addition to discrete tokens, to improve or maintain acoustic detail while reducing the computation involved in using hierarchical codecs.

## 5.2   Speech+Text LM

Speech+text LMs (Nachmani et al., 2024; Nguyen et al., 2025; Défossez et al., 2024) are SLMs that jointly model the distribution of speech and text $p(text, speech)$, and can therefore understand and generate both modalities. Such models often start with generative speech+text pre-training (Section 4.1), and can then be post-trained in a variety of ways.

A notable example of this approach is the Moshi model (Défossez et al., 2024), which is the first open-source speech-in-speech-out SLM-based dialogue system with real-time inference capability. Moshi takes time-aligned text and discrete speech representations as inputs and generates outputs in both text and speech forms. Moshi's text processing components—$\text{Enc}^{\text{txt}}()$, $\text{Adp}^{\text{txt}}()$, and $\text{Dec}^{\text{txt}}()$, and the LM backbone $\text{Seq}()$—are initialized from a pre-trained Helium text LLM (Défossez et al., 2024). After text pre-training, the LM is continually trained on joint prediction of the next speech and text tokens, followed by post-training in duplex mode (described in detail in Section 6) on conversation data. Moshi's training strategy produces a model that can generate consistent speech across long (several-minute) monologues and can engage in multi-turn conversations (Défossez et al., 2024).

Speech+text LMs vary widely in their decoding methods and approaches to constructing speech and text inputs/outputs. For example, SpiRit-LM (Nguyen et al., 2025) uses interleaved sequences of text and discrete speech tokens as both inputs and outputs, focusing solely on pre-training without doing any task-specific post-training. Compared to pure speech LMs, SpiRiT-LM has improved semantic understanding, as measured by the StoryCloze test (see Section 7.1 for more details). A precursor of this idea, SUTLM (Chou et al., 2023), modeled interleaved speech and text units of various sizes, but did not include a waveform synthesizer and was evaluated mainly on spoken language understanding tasks, with the main goal of enabling cross-modality transfer. Zeng et al. (2025) scale up the idea of interleaved speech-text training by using a speech synthesizer to generate massive datasets of interleaved speech-text sequences. As alternatives to time-aligned or interleaved speech and text sequences, SpeechGPT (Zhang et al., 2023a) outputs a sequence of text followed by corresponding speech, while Mini-Omni (Xie & Wu, 2024a;b) and LLaMA-Omni (Fang et al., 2025a) generate text and speech using separate output channels with a delay pattern in the speech token prediction (see Section 3.3.2). All three of these models (SpeechGPT, Mini-Omni, and LLaMA-Omni), unlike Moshi, focus on a traditional turn-taking structure and therefore have limited capability to model spontaneous turn-taking behavior (see Section 6 for more on conversation modeling).

## 5.3   Speech-aware text LM

Models in this category combine text LMs with speech encoders, and usually take both speech and text as inputs and generate responses in text form. The text input $X^{\text{txt}}$ can include instructions asking the model to perform tasks related to the speech input $X^{\text{sp}}$, possibly including tasks for which the model is not specifically trained. This type of SLM is typically initialized with a text LLM (which has often already been post-trained with instruction tuning or RLHF) so as to inherit its linguistic knowledge and instruction-

following capabilities. After combining the text LLM with a speech encoder and modality adapter, it is common to apply training methods that encourage better alignment between the speech and text modalities (Section 4.1.3) or to start post-training with an ASR task (Section 4.3) to help extract content information from speech inputs. The following post-training typically uses speech instruction-tuning datasets with diverse instructions (Section 4.2.2).

WavPrompt (Gao et al., 2022) represents the first example of this line of research. WavPrompt combines a wav2vec 2.0 (Baevski et al., 2020) SSL speech encoder with a GPT-2 text LM and is trained for speech classification tasks, keeping the LM backbone frozen and updating only the speech encoder. Although WavPrompt can improve over ASR+text model baselines on certain tasks, it was not evaluated on unseen tasks. Since then, many models in this category have been proposed.

SALMONN (Tang et al., 2024) is another early and impactful approach. The model is built on a pre-trained Vicuna LM backbone (Zheng et al., 2023). During post-training, the model is first trained on ASR and audio captioning and later on a more diverse set of speech understanding tasks. SALMONN's post-training keeps most of the parameters frozen—only the speech modality adapter and LoRA (Hu et al., 2022) adapters for the LM are learned—but the model is still able to generalize to unseen tasks with various natural language prompts.

Several other speech-aware text LMs were developed nearly simultaneously with SALMONN, facilitated by the release of open-source LLMs like LLaMA (Touvron et al., 2023a) and Vicuna (Zheng et al., 2023), often post-trained and evaluated on different tasks (Gong et al., 2023b; Wang et al., 2023b; Chu et al., 2023; Chen et al., 2024d). For example, LTU-AS (Gong et al., 2023b) combines spoken content understanding with paralinguistic analysis tasks (e.g., emotion recognition). SLM (Wang et al., 2023b) generalizes to unseen tasks specified by natural language prompts even while the LM backbone and speech encoder are both frozen, and only the speech adapter is optimized during instruction-based training, but focuses only on content-heavy tasks. Qwen-Audio (Chu et al., 2023) shares similar post-training tasks to those of SALMONN, but includes learning of the speech encoder and LM backbone (sequence model) weights. During post-training, Qwen-Audio starts with a more diverse task set instead of the single ASR task, including both content and paralinguistics, followed by a chat-based supervied fine-tuning stage that improves the model's robustness to variation in input prompts. The follow-up work, Qwen2-Audio (Chu et al., 2024) also applies direct preference optimization (Rafailov et al., 2023) after supervised fine-tuning. On the other hand, the more recent DeSTA (Lu et al., 2024) and DeSTA 2 (Lu et al., 2025a) propose a single descriptive speech-text alignment post-training task which requires the model to both recognize the spoken content and describe paralinguistic attributes.

Table 6 summarizes and compares a representative set of these models' training and evaluation choices. One key consideration is the tradeoff between preserving knowledge acquired through text pre-training and learning new speech tasks, especially those requiring non-textual information such as speech emotion. An important design choice, therefore, is which of the pre-trained parameters to freeze or update, and this issue has not yet been thoroughly explored.

In addition to the multiple post-training approaches, speech-aware text LMs also include some differences in architecture choices. For example, WavLLM (Hu et al., 2024) combines both SSL and supervised pre-trained speech encoders, and BESTOW (Chen et al., 2024e) experiments with a sequence-to-sequence model in addition to the common decoder-only LM architecture. While most models in this category are initialized with pre-trained text LLMs, UniverSLU (Arora et al., 2024) starts from a Whisper model for conditional pre-training (see Section 4.1.2) and is then fine-tuned for various understanding tasks, first by prepending new task specifiers and then by adding natural language instructions to the input.

# 6 Duplex speech dialogue

As discussed in the previous sections, most SLMs to date assume that dialogue with a user follows a turn-by-turn structure, where a user provides an input and the SLM generates a response. However, natural speech dialogue is *duplex*: Both speakers can simultaneously send and receive information. Compared to text-based interaction, duplex dialogue presents unique challenges. Figure 6 (a) shows an example of a full-duplex

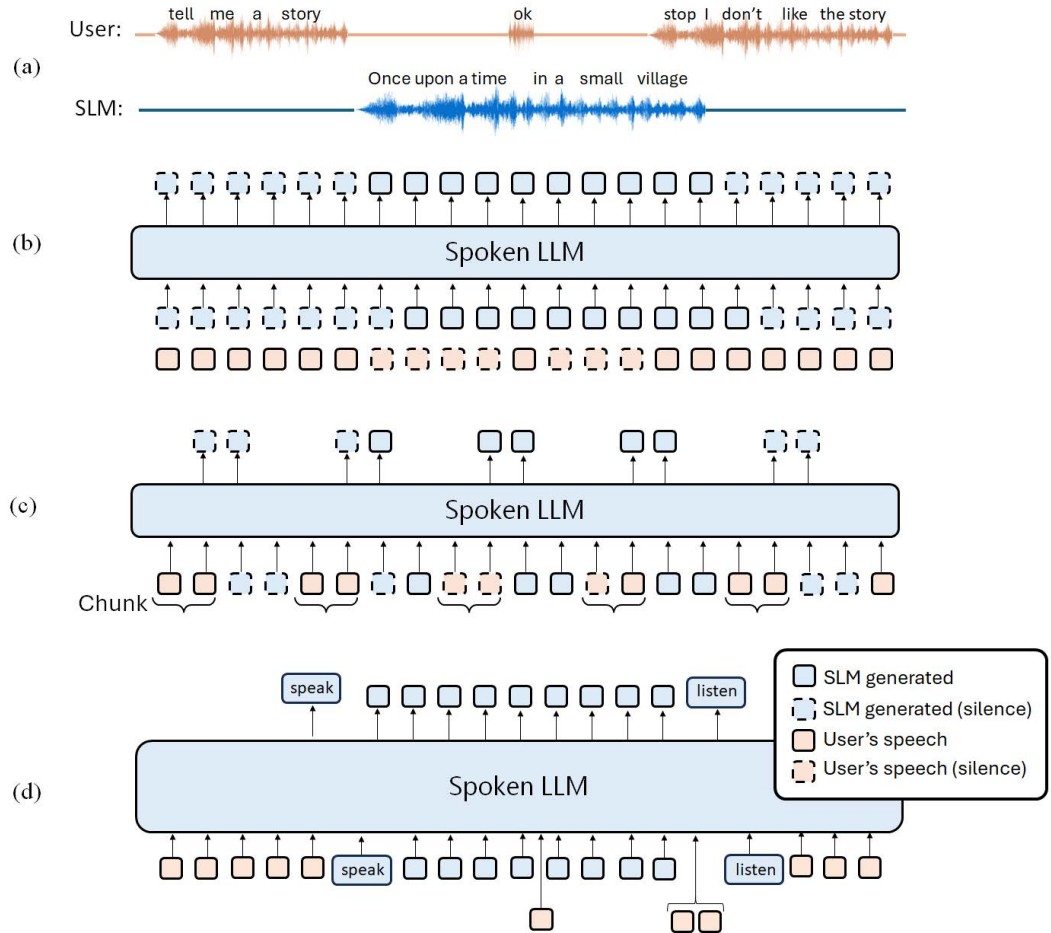

Figure 6: Full-duplex speech conversation. (a) An example of full-duplex speech conversation between a user and an SLM. (b) Dual-channel approach. (c) Time-multiplexing approach (with equal chunks). (d) Time-multiplexing approach (where the SLM controls the switching between listening and speaking modes).

dialogue. The SLM must determine whether the user has finished their utterance before starting to speak. Backchannel signals (e.g., saying "mm-hmm," "I see," or "okay") and non-verbal vocalization (e.g., laughter) are often used to facilitate smoother conversation. Additionally, the SLM may be interrupted while talking and must respond appropriately to maintain the flow of the dialogue. None of these characteristics exist in text-based dialogue. Over the years, researchers have explored various methods to enable duplex dialogue in speech systems (Masumura et al., 2018; Roddy et al., 2018; Skantze, 2017; Meena et al., 2014; Ekstedt & Skantze, 2020).[8] Rather than reviewing all prior work on spoken duplex dialogue, we focus specifically on end-to-end SLM approaches.

**Dual channel** One way to achieve full-duplex dialogue is by using a dual-channel approach (Nguyen et al., 2023; Défossez et al., 2024; Ma et al., 2025; Meng et al., 2024), as shown in Figure 6 (b). The SLM has two input channels: the listening channel (the sequence with red blocks) which continuously receives input, and the speaking channel (the sequence with blue blocks) where the spoken output from the SLM is directed, allowing the model to track what it has said. The model produces output at each step, including tokens representing speech or silence (blocks with a dotted outline). In this way, an SLM can listen to the user's words while generating output simultaneously. An early representative model for this method is the dialogue GSLM (dGSLM) (Nguyen et al., 2023); Moshi has also implemented this strategy (Défossez

---

[8]There is an extensive amount of older related literature, including on rule-based dialogue modeling; here, we only cite a few machine learning-based approaches.

et al., 2024). One challenge when using the dual-channel approach is that, instead of a typical autoregressive model, a specialized architecture is required. dGSLM (Nguyen et al., 2023) uses a dual-tower transformer architecture, where two transformers handle the two channels, using cross-attention to exchange information. Other models (Défossez et al., 2024; Ma et al., 2025; Meng et al., 2024) modify the input structure of the transformer to accommodate the dual-channel input.

**Time multiplexing**  Figure 6 (c) and (d) illustrate the time multiplexing approach. In this approach, the SLM has only one channel, so it must switch between listening and speaking modes. During the listening mode, the SLM takes input from the user (red blocks) without generating output. During the speaking mode, the SLM generates speech representations (blue blocks) and takes its own output as input in an autoregressive manner. The strength of this approach is that the sequence model can be a typical decoder-only autoregressive model, and can therefore be initialized with a text LLM.

The core challenge is determining how to switch between listening and speaking modes. One approach, shown in Figure 6 (c), is to alternate between processing a fixed-duration time slice from the user's input and then switching to the speaking mode (Zhang et al., 2024c). This approach has been adopted in Synchronous LLM (Veluri et al., 2024), OmniFlatten (Zhang et al., 2025), and SCoT Arora et al. (2025c).

Another approach allows the model to determine when to switch modes (Wang et al., 2024b; Xu et al., 2024; Wang et al., 2025b), as shown in Figure 6 (d). In listening mode, while processing the user's input, at each time step the model predicts whether to initiate a response. This decision is signaled by the prediction of a special token ([speak], as shown in the figure). The model processes its generated output autoregressively until it produces the [listen] token, which triggers a switch back to listening mode. Users can continue providing input during speaking mode, dynamically influencing the model's responses. In particular, user input influences the [listen] token generation, enabling users to interrupt the model's output when desired. This interaction is implemented by interleaving the model's responses with user input, using some special tokens or embeddings to distinguish between the two sources.[9]

The architectural designs described above, whether dual-channel or time multiplexing, share a common goal: to enable natural user interaction. Evaluating how well a model has achieved this goal can be challenging, since most benchmarks involve offline rather than interactive evaluation. Section 7.3 below describes the community's efforts so far to benchmark interactivity in SLMs for dialogue.

# 7 Benchmarking and Evaluating SLMs

Like text language models, SLMs may possess a broad array of capabilities. In addition (and in contrast to text LMs), they also involve a variety of design choices that differ significantly across models. These properties can make it difficult to compare SLMs, and also make it more important to assess them from multiple angles and at multiple training stages. Below, we outline commonly used evaluation methods and benchmarks, categorized into four primary groups: (1) likelihood-based evaluation metrics; (2) generative metrics; (3) evaluation of interactivity; and (4) trustworthiness evaluations.

Traditional tasks (e.g., ASR, TTS, ST) are also frequently used to assess SLM performance. However, we exclude them from this survey, as they are well-established and extensively covered in the existing literature. For a more comprehensive survey of benchmarks for SLMs, please refer to a recent overview paper focusing on SLM benchmarks (Yang et al., 2025b).

## 7.1 Likelihood-Based Evaluation

Likelihood-based evaluation metrics (Dunbar et al., 2021) are designed to assess the sequence modeling capabilities of SLMs, typically for the pre-training stage of pure speech LMs and joint speech+text LMs (Lakhotia et al., 2021; Hassid et al., 2023; Borsos et al., 2023a; Défossez et al., 2024). Such evaluation metrics can be considered analogues to perplexity (PPL) for text LMs. Since there is no standard speech tokenization,

---

[9]Theoretically, there is always some input, even when no one is speaking. Here, we assume that a voice activity detection (VAD) system is in place, so only the tokens representing actual user speech are sent to the SLM during speaking mode.

the speech "vocabulary" may change between pure SLMs. Therefore, PPL cannot be directly applied. Like perplexity, likelihood-based metrics are viewed not as tasks, but rather as proxies indicating the ability of the base SLM to represent the speech distribution it is modeling. The assumption is that good performance on these metrics will, after fine-tuning, be reflected in improved downstream task performance, and there are some indications from analyses of likelihood-based metrics that this assumption holds (Dunbar et al., 2022).

We emphasize that speech tokenization plays a significant role in shaping sequence modeling capabilities and therefore cannot be disregarded. For instance, utilizing tokens generated by a self-supervised learning model such as HuBERT significantly outperforms the quantization of log mel spectrograms (Lakhotia et al., 2021). Likewise, employing speech tokenizers with higher compression rates (i.e., sampled at lower frequency in the latent space) yields better performance in speech modeling (Borsos et al., 2023a; Hassid et al., 2023). While the likelihood-based metrics in this subsection focuses on the sequence model, in principle the two components should be evaluated together.

In likelihood-based evaluations, the pre-trained SLM is typically provided with two input sequences: one representing a natural speech sequence (positive example) and the other an unnatural sequence relative to a specific property (negative example). We then calculate the percentage of instances where the SLM assigns a higher likelihood to the positive example than to the negative one. This type of evaluation has also been popular in the NLP community until very recently (Zellers et al., 2019; Mostafazadeh et al., 2016; Touvron et al., 2023b; Gemma Team et al., 2024; LCM team et al., 2024). Due to significant model improvements, this type of NLP benchmark has saturated and significantly more complex ones have emerged (Jimenez et al., 2024; Srivastava et al., 2023).

Metrics in this category are designed to evaluate a model's capacity to represent various speech characteristics. For example, sWUGGY (Dunbar et al., 2021) evaluates the lexical capabilities of models by presenting them with pairs of utterances, one consisting of a real word and the other a similar non-word (e.g., 'brick' vs. 'blick'), where nonwords were obtained from the text Wuggy task (Keuleers & Brysbaert, 2010). Similarly, sBLIMP (Dunbar et al., 2021) (which was adapted from the text-based BLiMP (Warstadt et al., 2020)) evaluates the ability of the model to represent syntactic properties. The model is presented with a matched pair of grammatically correct and incorrect sentences. Hassid et al. (2023) expanded this method to evaluate semantic understanding of spoken text. They generated two spoken adaptations of the StoryCloze text paragraph story completion task (Mostafazadeh et al., 2016). In the first adaptation, they adhered to the original StoryCloze setup, while in the second, they randomly sampled negative examples, resulting in a simplified version primarily requiring an understanding of the general topic of the paragraph. ProsAudit (de Seyssel et al., 2023) constructs negative examples by inserting pauses in unnatural positions in utterances to evaluate SLMs' sensitivity to speech prosody. Lastly, Maimon et al. (2025b) proposed the SALMon evaluation suite, which, unlike the previously mentioned benchmarks, focuses on a variety of non-lexical acoustic and prosodic elements (e.g., speaker identity, speaker sentiment, background noise, and room acoustics) at two levels of complexity: One measures the *consistency* across time of a given acoustic element, whereas the other measures *alignment* between the acoustic details and the semantic, spoken content.

These likelihood-based metrics evaluate complementary aspects of SLM performance, and performance across benchmarks is not necessarily correlated. For example, even though some SLMs manage to model sentiment to some extent (as shown by the SALMon sentiment consistency measure) and have good text modeling capabilities (as demonstrated by their performance on sWUGGY), they achieve near-random performance on the SALMon alignment measure (Maimon et al., 2025b). This shows that jointly reasoning over both text and acoustic content is still challenging for SLMs. Similarly, some recent studies on newly proposed SLMs (Tseng et al., 2025; Chou et al., 2025) show that the models achieving the best performance on content modeling (e.g., on StoryCloze or sWUGGY) are not always the best at handling acoustic details (as indicated by their performance on SALMon).

## 7.2 Generative Metrics

Unlike likelihood-based metrics, which assess a model's ability to assign higher likelihoods to in-domain samples than out-of-domain ones, generative metrics focus on evaluating the model's ability to produce

meaningful continuations or responses based on a given prompt or instruction. Such evaluation methods are suitable for all SLM types.

**Intrinsic quality metrics**  For SLMs that generate spoken output (e.g., speech+text LMs), the quality of the generated response can be evaluated along several axes:

1. Speech quality, using subjective tests or objective metrics such as MOSNet (Lo et al., 2019). While this is primarily a speech synthesis metric that emphasizes signal quality, it can also be influenced to some degree by the intelligibility of the generated speech (Lakhotia et al., 2021)

2. Speaker, acoustic, and/or sentiment consistency using human judges or pre-trained classifiers (Nguyen et al., 2025). This group of metrics measures how consistent the generated speech is with the input speech prompt, in terms of different acoustic properties.

3. Quality of the spoken content, evaluated by either comparing to a target response or transcribing the speech and using a text LLM as a judge to score the content (Arora et al., 2025c; Zhang et al., 2025).

Like perplexity-based metrics, this set of evaluation measures is not tied to any specific downstream task. Instead, their primary purpose is to evaluate the generative quality of SLMs along various dimensions.

For pure SLMs and speech+text SLMs, a common evaluation metric is the *generative perplexity*, in which we transcribe the generated speech and measure its perplexity (PPL) using a pre-trained text LLM (Lakhotia et al., 2021). However, a known problem with this metric is that text LLMs tend to assign high probability to repeated content (Holtzman et al., 2020). To mitigate this issue, Lakhotia et al. (2021) propose the VERT metric, which balances between quality and diversity of the generated response. VERT uses a linear combination of the text PPL and the auto-BLEU metric of within-sentence diversity (Lakhotia et al., 2021).

**Task-based evaluation**  For joint speech+text LMs and speech-aware text LMs, a common approach is to evaluate the model's ability to answer questions and follow instructions. Nachmani et al. (2024) proposed two spoken question answering benchmarks: LLaMA-Questions, derived from a text LLM, and a synthesized version of the WebQuestions (Berant et al., 2013) textual benchmark. Défossez et al. (2024) built upon this approach and developed a synthesized version of TriviaQA (Joshi et al., 2017). To evaluate SLM responses on these question-answering benchmarks, the generated speech is transcribed and the accuracy is measured against the target answer. Chen et al. (2024c) proposed VoiceBench, an extended question-answering dataset consisting of (1) open-ended QA; (2) reference-based QA; (3) multiple-choice QA; and (4) instruction-following evaluations. VoiceBench also evaluates the model's ability to handle background noise and speaker variability. Fang et al. (2025a) proposed to evaluate the instruction-following capabilities of SLMs using *LLM-as-a-judge* methods. In this approach, an external text LLM rates the quality of the responses considering both content and style.

Recently, there have been several efforts to scale up evaluations of speech-aware text LMs to a broader range of tasks and instruction data. In these evaluations, each task consists of text instructions, speech utterances, and text labels. Huang et al. (2024) introduced the Dynamic-SUPERB evaluation suite, designed as a dynamic benchmark allowing for community contributions of tasks, analogously to BIG-bench (Srivastava et al., 2023) for text LMs. The initial phase of Dynamic-SUPERB focused on classification tasks related to content, speaker characteristics, semantics, degradation, paralinguistics, and non-speech audio information. Phase 2 of Dynamic-SUPERB (Huang et al., 2025) significantly expanded the task set to include regression and sequence generation tasks, making it the largest benchmark for speech and audio evaluation. Yang et al. (2024b) proposed AIR-Bench, which includes both classification and open-ended chat-style questions. Both Dynamic-SUPERB Phase-2 and AIR-Bench use LLM-as-a-judge evaluation. AudioBench (Wang et al., 2025a) includes eight speech and audio tasks based on 26 datasets, focusing on speech understanding (e.g., ASR, spoken QA, speech instruction), audio scene comprehension (e.g., audio captioning, audio scene question answering), and voice analysis (e.g., accent recognition, gender recognition, emotion recognition). Sakshi et al. (2025) introduced MMAU, a benchmark for audio understanding and reasoning using natural language. MMAU includes both information extraction tasks (e.g., "Which word appears first?") and reasoning tasks

(e.g., "What are the roles of the first and second speakers in the conversation?"), formulated as multiple-choice questions. When we compare model performance on MMAU and VoiceBench, an interesting trend emerges: Their results are negatively correlated.[10] VoiceBench emphasizes QA ability, which is primarily content-oriented, whereas MMAU covers a broader range of tasks and perspectives. This divergence underscores the necessity of evaluating SLMs from multiple angles, rather than relying on a single benchmark, to obtain a comprehensive understanding of their capabilities.

**Conversation-based evaluation**   While the benchmarks above cover a wide variety of tasks, some recent ones focus specifically on whether SLMs can perceive acoustic details in conversations and respond appropriately. For example, StyleTalk (Lin et al., 2024a) assesses SLMs' ability to generate text responses that align with specific speaking styles, for example in terms of intonation and emotion. The evaluation compares the generated output with a target response using standard text generation metrics (BLEU, ROUGE, and others). Similarly, the E-chat200 dataset (Xue et al., 2024) also assesses a model's ability to generate responses that align with the speaker's emotion. The dataset was synthetically created using an expressive TTS, while the text-based questions and responses were generated by an external LLM. SD-Eval (Ao et al., 2024) and VoxDialogue (Cheng et al., 2025) go beyond emotion alignment to include alignment with other speaker and style features (e.g., accent, age, prosody, timbre, volume) as well as environmental context. The recently introduced StressTest benchmark (Yosha et al., 2025) evaluates the ability of speech-aware SLMs to understand differences in meaning conveyed by sentence stress, finding that existing models tend to do poorly on this task. VoxRole (Wu et al., 2025d) is a benchmark designed for speech-based role-playing conversational agents, which evaluates how well they maintain their personas. S2S-Arena (Jiang et al., 2025) evaluates SLMs' ability to respond appropriately to spoken instructions that involve paralinguistics. Unlike the benchmarks above that rely on automatic evaluation, S2S-Arena uses human judgments to evaluate SLMs.

**Comparisons of cascaded vs. end-to-end SLMs**   When evaluating SLMs, researchers often include cascaded baselines for comparison with end-to-end models. For speech-aware text LMs, many studies contrast end-to-end approaches with ASR → LLM cascaded systems. Across benchmarks, cascaded systems typically excel at semantic tasks such as spoken language understanding but underperform on speaker-related and paralinguistic tasks (Huang et al., 2024; 2025; Sakshi et al., 2025; Liu et al., 2025; Yan et al., 2025; Yang et al., 2025a). While cascaded systems remain strong baselines, jointly optimizing the speech encoder with the LLM can yield additional benefits by better aligning speech representations with the backbone sequence model. These benefits include improved instruction following (Lu et al., 2025c) and paralinguistic understanding (Lu et al., 2025b). However, such joint training also risks catastrophic forgetting when integrating speech encoders and pre-trained sequence models (Lu et al., 2025c; Hsiao et al., 2025).

When considering SLMs with full-duplex behaviors, trade-offs between cascaded and end-to-end models become even clearer. Cascaded approaches (typically ASR → LLM → TTS) provide modular flexibility and strong semantic modeling (Wang et al., 2025b; Zeghidour et al., 2025), but they often incur higher latency and lose fine-grained conversational behavior modeling (Lin et al., 2025b; Arora et al., 2025b; Chang et al., 2025). End-to-end models can mitigate some of these limitations by enabling smoother turn-taking and backchanneling behaviors (Arora et al., 2025b; Lin et al., 2025b). However, challenges remain regarding their overall modeling capacity, as they often rely on fine-tuning a text-based LLM to directly model both the listening and speaking channels (Défossez et al., 2024), as discussed in Section 6, which is inherently difficult. Therefore, developing robust end-to-end architectures for full-duplex SLMs remains a major challenge (Lin et al., 2025b). For a broader discussion of the trade-offs between cascaded and end-to-end SLMs, readers may refer to the recent survey on SLM evaluation by Yang et al. (2025b).

---

[10]This comparison was conducted on Oct 3rd, 2025. The results of VoiceBench are from `https://github.com/MatthewCYM/VoiceBench`, while the results of MMAU are from `https://sakshi113.github.io/mmau_homepage/#leaderboard-v15-parsed`, using MMAU-v05.15.25. We evaluated 6 models on both benchmarks. The metrics are the overall score for VoiceBench and the average test score for MMAU. We found that the correlation coefficient between these scores is negative.

### 7.3 Evaluating Interactivity

When evaluating SLMs for spoken dialogue (described in Section 6), quality is in large part determined by *interactivity* measures; that is, how well a model handles the dynamics of a real conversation, such as engaging in smooth turn-taking, providing timely backchannels, and responding gracefully to user interruptions (barge-in). Evaluating these complex dynamics has progressed from early statistical analyses to modern automated benchmarks and subjective user studies.

Pioneering work in the dialogue SLM area, such as dGSLM (Nguyen et al., 2023), introduced methods for evaluating dialogue by identifying basic conversational patterns and comparing model-generated dialogue with ground-truth human conversations. dGSLM's evaluation was based on several key components of turn-taking derived from voice-activity–based statistics (e.g., inter-pausal units, pauses, gaps, and overlaps) computed from human–human interactions.

One of the most basic components of SLMs' interactive capabilities is *latency*, typically evaluated by the delay from when the user stops speaking to when the model begins its spoken response, often termed *first-packet latency* (Yan et al., 2025; Xu et al., 2025) or *response latency* (Wang et al., 2024b; Lin et al., 2025b). The average human conversational response latency has been estimated at approximately 200 ms (Stivers et al., 2009). Latency reported for dialogue models depends greatly on hyperparameters, hardware specifications, and network conditions. Table 3 lists a selection of models and their reported latencies from the recent literature.

Recent efforts to evaluate interactivity have moved beyond basic conversational statistics or latency toward automated, scenario-driven benchmarks, which provide more user-oriented measures of interactive performance. For example, Full-Duplex-Bench (Lin et al., 2025b;a) assesses full-duplex behaviors including pause management, backchannel appropriateness, smooth turn transitions, and interruption handling. The more recent Game-Time Benchmark (Chang et al., 2025) evaluates the temporal dynamics of SLMs, specifically testing their time awareness, adherence to tempo, and ability to engage in simultaneous speech with users.

While automated benchmarks like Full-Duplex-Bench and the Game-Time Benchmark provide useful quantitative and qualitative evaluation, user studies remain essential for capturing interaction quality. For instance, "Talking Turns" (Arora et al., 2025b) compared the practical behaviors of a VAD-based cascaded system against an end-to-end model, Moshi Défossez et al. (2024). This study found that while the VAD-based system was better at yielding its turn when interrupted, the two systems shared some weaknesses such as rarely producing backchannels and often failing to take a turn when a user expected them to. A combination of automated benchmarks and human-in-the-loop evaluations will likely continue to be necessary to advance toward truly natural conversation flow.

### 7.4 Evaluating Trustworthiness

The benchmarks discussed thus far focus on the *performance* of SLMs. Next, we consider evaluations that measure their *trustworthiness*. Since SLMs generate word sequences (whether directly as text or within the spoken output), all trustworthiness considerations related to text-based LLMs also apply to SLMs. However, in addition to this content information, non-content information in the generated speech creates additional aspects of trustworthiness that need to be considered.

#### 7.4.1 Hallucination

Text-based LLMs often suffer from hallucinations, or outputs that are inconsistent with the given context or other knowledge considered to be fact. In addition to such content hallucinations, SLMs may also generate audio hallucinations. For example, consider an audio clip without a dog barking. If you ask an SLM to describe it, it would most likely not mention anything related to a dog barking. However, if you directly ask the model, "Is there a dog barking?" it might answer, "Yes," even though it can provide accurate audio captions when explicitly prompted to do so (Kuan & Lee, 2025).

Kuan & Lee (2025) study whether SLMs can accurately perform three types of tasks without hallucination: (1) identify the presence of an object (e.g., "Is there a dog barking?"), (2) determine the temporal order of

Table 3: Latency comparison of speech-text models, measured in milliseconds (ms). Official claims are denoted separately from benchmarked results reported by URO-Bench (Yan et al., 2025) and LLaMA-Omni2 (Fang et al., 2025b).

| Model | Size (# params) | Latency (ms) | Device |
|---|---|---|---|
| Moshi | 7B | 200 | Official claim |
| GPT-4o | – | 320 | Official claim |
| Qwen3-Omni-30B-A3B | 34B | 234 | Official claim |
| *Latency reported from URO-Bench* | | | |
| Freeze-Omni | 7B | 3675 | A40 |
| Mini-Omni | 0.5B | 399 | A40 |
| Mini-Omni2 | 0.5B | 402 | A40 |
| GLM-4-Voice | 9B | 3244 | A40 |
| *Latency reported from LLaMA-Omni2* | | | |
| SpeechGPT | 13B | 5588 | L40 |
| GLM-4-Voice | 9B | 1563 | L40 |
| LLaMA-Omni | 8B | 347 | L40 |
| LLaMA-Omni2-0.5B | 0.5B | 543 | L40 |
| LLaMA-Omni2-14B | 14B | 663 | L40 |

sound events (e.g., "Is the dog barking before someone laughs?"), and (3) recognize the source of a sound event (e.g., "Who is laughing, a man or a woman?"). The main finding is that all of the evaluated SLMs hallucinate more than a simple cascade model that combines audio captioning with text LLMs. This issue may arise because, while SLMs can generate accurate audio captions, they struggle to understand specific questions Kuan et al. (2024). The authors also proposed a method that mitigates hallucination by prompting the model to produce output in multiple steps (similarly to chain-of-thought prompting (Wei et al., 2022b)), by first describing the audio and then responding to the instruction based on that description.

### 7.4.2 Toxicity

Prevention of harmful, offensive, or inappropriate output is a crucial issue for text LLMs, and the same challenge also applies to SLMs. Both Meta's SpiRit-LM (Nguyen et al., 2025) and OpenAI's GPT-4o voice mode (OpenAI, 2024) evaluate the toxicity of their models' responses. These evaluations typically involve using specific prompts to elicit speech responses from the SLMs and assessing the toxicity of the text transcriptions of those responses. These evaluations therefore consider only verbal toxicity, i.e. toxicity within the word sequence. The evaluation of non-verbal toxicity (e.g., toxic sarcasm) has yet to be widely studied.

### 7.4.3 Bias

Similarly to toxicity, all bias measures used for text LLMs can also be applied to SLMs. For SLMs that produce spoken output, text-based methods can still be used to analyze their transcriptions (Lin et al., 2024c). In addition, because speech conveys information beyond the textual content, there may be factors outside the text that indicate bias. Lin et al. (2024b) investigate such biases by providing an SLM with input utterances that share identical content but differ in speaker characteristics (such as gender and age) to assess whether these traits influence its responses. The results suggest that the tested SLMs exhibit minimal bias. However, this may be because current models lack sufficient sophistication to recognize differences in speaker characteristics, and therefore do not respond differently based on them. OpenAI reported in a blog post (OpenAI, 2024) on their efforts to ensure consistency across user voices by post-training GPT-4o with a diverse set of voices and on a bias evaluation, which found no significant variation in model behavior across

voices. While such findings are encouraging, regular and thorough bias benchmarking is needed to track the evolution of SLM bias as this field matures.

### 7.4.4 Deepfakes

Current SLMs can mimic a variety of voices to a level indistinguishable from human speech (Chen et al., 2025b). Unfortunately, malicious actors may exploit this technology, leading to misuse and security concerns, such as deepfake attacks (for example, creating fake news using the voice of a public figure). To address this issue, (Wu et al., 2024d; Du et al., 2025) introduced CodecFake, a deepfake audio dataset, and found that detection models trained on CodecFake can counter fake audio generated by existing SLMs.

## 8 Challenges and future work

This survey has covered some of the key milestones that have been achieved in SLM research: the first pure speech language models that could generate convincing-sounding stretches of English speech (GSLM (Lakhotia et al., 2021)); the first generation of models that could perform a variety of tasks with reasonable accuracy given natural language instructions (Gao et al., 2022; Gong et al., 2023b; Wang et al., 2023b); models that can converse in "full-duplex" mode (Défossez et al., 2024); and benchmarks tailored for evaluating the modeling and instruction-following capabilities of SLMs (Dunbar et al., 2021; Huang et al., 2025).

While research in this area has produced many SLMs and exciting outcomes, it is safe to say that we are not yet close to the goal of truly universal speech processing systems. This section highlights several categories of challenges and open questions, which suggest directions for future research.

**Model architecture** The optimal representation of speech within SLMs remains unclear. Speech representations in SLMs include both discrete and continuous varieties, derived from a wide range of encoders. This design choice can also influence other architectural choices in an SLM, for example depending on the information rate of the encoder and whether it encodes more phonetic or other types of information.

Another open question is determining the best method to combine speech and text, which applies to all aspects of SLM modeling and training. We have described various choices of modality adapters and approaches for interleaving speech and text. These have not been thoroughly compared, so the effect of each modeling choice is still unclear.

A final architectural challenge is that most current SLMs are large and slow, making them impractical for real-time and on-device settings. To some extent this is because various compression algorithms (e.g., (Lai et al., 2021; Peng et al., 2023a; Ding et al., 2024)) and alternative architectures (e.g., Park et al. (2025)) have not been widely applied to SLMs. However, there is also an inherent efficiency challenge that arises when combining multiple pre-trained components, sometimes with different architectures and frame rates.

**Training** There is a lack of public high-quality training data, particularly for instruction tuning and chat-based training. Zeng et al. (2025) showed that scaling synthetic data generation enhances the performance of pure speech LMs; similar research efforts could be directed toward instruction tuning and chat-based spoken data. Additionally, SLMs are trained on diverse datasets (including some proprietary datasets), making it difficult to analyze whether performance differences are caused by model design choices or training data. Thorough ablation studies focusing on the various design choices (Section 3) and training strategies (Section 4) are also essential and still lacking. Finally, scaling studies for SLMs are needed in order to better understand how SLMs scale with model and data size. Such studies would hopefully produce new scaling laws and help to speed up the development cycle. Cuervo & Marxer (2024) conducted the first such investigation for pure SLMs, and more recently Maimon et al. (2025a) empirically analyzed the optimization process for pure speech LMs, showing how one can train a high-quality pure SLM in 24 hours using a single GPU. It remains uncertain whether scaling findings apply also to speech+text LMs or speech-aware text LMs.

**Evaluation** Thus far, different SLMs have typically been evaluated on different datasets and tasks. Recent efforts to collect large numbers of tasks and standardize benchmarking (Huang et al., 2024; Yang et al.,

2024b; Huang et al., 2025) are promising, but as of this writing they are not yet widely adopted for evaluating new models. Existing benchmarks also do not cover the full range of spoken language tasks. The largest current benchmarking effort, Dynamic-SUPERB Phase-2 (Huang et al., 2025), includes 180 tasks, which is similar to the size of the BIG-Bench suite of text LLM tasks (Srivastava et al., 2023). However, the range of spoken language tasks is arguably much larger than that of text tasks, since speech tasks include the vast majority of text tasks (the ones that don't relate explicitly to the textual form, such as transliteration) and in addition include a variety of speech-specific tasks related to speaker properties, accents, or prosody-specific content. In addition, there is a lack of standardized benchmarking for speech *generation* tasks. Finally, there is a need for more benchmarking of latency and turn-taking behavior in conversational SLMs (although, as mentioned in earlier sections, some benchmarks have begun to address these dimensions (Yan et al., 2025; Lin et al., 2025b)).

**Open research**  Very few SLMs are fully open-source—including code, model checkpoints, and training data—which makes a comprehensive comparison between approaches virtually impossible. There has been progress in this direction, with some models having at least publicly available weights (see Figure 1), and some support in open-source toolkits (Tian et al., 2025b). Many of the items on the wish list above, such as controlled comparisons between multiple design decisions, will be infeasible to accomplish until more fully open models are released.

**Improving inclusiveness and safety**  SLM research has, thus far, understandably focused on high-resource languages and settings. As SLMs become more performant and enter commercial products in daily use, it will be critical to make them accessible to as broad a range of users as possible, including a variety of languages, dialects, and speech-related medical conditions. Research in this area will likely follow the arc of text LLM research, but again, there are speech-specific challenges that arise from speaker and style variation. Similarly, the area of safety and trustworthiness has only begun to be explored, and will require speech-specific solutions to speech-specific challenges like deepfakes and speaker type-related bias.

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

# A   Appendix

Table 4: Selected spoken language models. *P.*: Phonetic token. *A.*: Acoustic token. *C.*: Continuous feature. For components not explicitly cited in the table, please see the following references: SSE (Algayres et al., 2022), USM (Zhang et al., 2023b), SNAC (Siuzdak et al., 2024), SenseVoice (An et al., 2024), and Canary (Puvvada et al., 2024) for speech representation models, and Tacotron 2 (Shen et al., 2018), Wave-Fit (Koizumi et al., 2023), and CosyVoice (Du et al., 2024) for speech decoders. Modalities other than text or speech (e.g., vision) are omitted. [†]: Pre-trained ASR encoder.

| Model | Input Representation → Output Representation | Speech Decoder | Modality Adapter | Description |
|---|---|---|---|---|
| *Pure Speech LM* | | | | |
| GSLM (Lakhotia et al., 2021) | HuBERT (*P.*) | Tacotron 2 | X | The first pure speech LM, and the first attempt at using phonetic tokens as basic units for language modeling. |
| pGSLM (Kharitonov et al., 2022b) | HuBERT (*P.*) + F0 + duration | HiFi-GAN | X | Augments GSLM with discrete prosodic tokens. |
| dGSLM (Nguyen et al., 2023) | HuBERT (*P.*) + duration | HiFi-GAN | X | The first speech LM to model human conversation end-to-end (but not a full-duplex model, as it does not take environmental audio input). |
| tGSLM (Algayres et al., 2023) | SSE (*P.*) + HuBERT (*P.*) | Tacotron 2 | X | Reduces sequence length and improves memory efficiency of GSLM by using word-sized basic units. |
| AudioLM (Borsos et al., 2023a) | w2v-BERT (*P.*) → SoundStream (*A.*) | SoundStream | X | The first work to model audio codec units with a language model, and also introduces the coarse-to-fine generation pattern in Figure 5(a). |
| TWIST (Hassid et al., 2023) | HuBERT (*P.*) | HiFi-GAN | X | Demonstrates the benefits of initializing a speech LM with a pre-trained text LM. |
| SoundStorm (Borsos et al., 2023b) | w2v-BERT (*P.*) → SoundStream (*A.*) | SoundStream | X | Improves the efficiency of coarse-to-fine generation in AudioLM by using a non-autoregressive model to generate the fine-grained tokens. |
| Align-SLM (Lin et al., 2025c) | HuBERT(*P.*) | HiFi-GAN | X | First work to align pure speech LM generations with reinforcement learning feedback. |
| *Speech+Text LM* | | | | |
| Moshi (Défossez et al., 2024) | Text + Mimi (*A.*) | Mimi | Vocabulary Expansion | The first full-duplex speech language model, and proposes the decoding pattern in Figure 5(c). |
| GLM-4-Voice (Zeng et al., 2024) | Text + Whisper[†] (*P.*) | CosyVoice | Vocabulary Expansion | Demonstrates the capabilities of fixed-ratio text-speech token interleaved prediction, without forced alignment. |
| SpiRit-LM (Nguyen et al., 2025) | Text + HuBERT (*P.*) | HiFi-GAN | Vocabulary Expansion | Demonstrates the decoding pattern in Figure 5(d). |
| MinMo (Chen et al., 2025a) | Text + SenseVoice[†] (*C.*) → Text + SenseVoice[†] (*P.*) | CosyVoice | Transformer + CNN | A full-duplex model trained with a multi-stage alignment strategy, including a newly proposed method using text model representations as context for the speech decoder. |
| VoxtLM (Maiti et al., 2024) | Text + HuBERT (*P.*) | HiFi-GAN | Vocabulary Expansion | Integrates text and speech modalities into one decoder-only model. |
| AnyGPT (Zhan et al., 2024) | Text + SpeechTokenizer (*A.*) | SpeechTokenizer | Vocabulary Expansion | An any-to-any multimodal language model, with various encoders and decoders (similar to SpeechGPT (Zhang et al., 2023a)). |
| MiniOmni (Xie & Wu, 2024a) | Text + Whisper[†] (*C.*) → Text + SNAC (*A.*) | SNAC | MLP & Vocabulary Expansion | Demonstrates the decoding pattern in Figure 5(a). |
| LLaMA-Omni (Fang et al., 2025a) | Whisper (*C.*) → Text + HuBERT (*P.*) | HiFi-GAN | MLP | Demonstrates the decoding pattern in Figure 5(b). |
| SLAM-Omni (Chen et al., 2025c) | Text + Whisper[†] (*P.*) | CosyVoice | Linear | Proposes an approach that decodes one text token and several audio tokens simultanously. |
| Spectron (Nachmani et al., 2024) | Text + Mel-Spectrogram (*C.*) | WaveFit | MLP | An LM that directly models spectrograms. |
| OpusLM (Tian et al., 2025a) | Text + XEUS (*P.*)+ ESPnet-Codec(*A.*) | ESPnet-Codec | Vocabulary Expansion | Family of open-source SLMs with competitive performance to closed-source APIs on speech and text processing tasks. |

Table 5: Selected spoken language models (continued).

| Model | Input Representation → Output Representation | Speech Decoder | Modality Adapter | Description |
|---|---|---|---|---|
| | | *Speech-Aware Text LM* | | |
| WavPrompt (Gao et al., 2022) | Text + Wav2vec 2.0 (*C.*) → Text | X | CNN | The first speech-aware text LM. Requires few-shot demonstration. |
| X-LLM (Chen et al., 2023) | Text + Conformer[†] (*C.*) → Text | X | CIF + Transformer | One of the earliest multimodal LLMs that include speech input. Combines a frozen LLM (ChatGLM) with frozen speech (and image) encoders, and trains modality adapters only. |
| LTU-AS (Gong et al., 2023b) | Text + Whisper[†] (*C.*) → Text | X | TLTR | Proposes an approach for universal audio perception by training on QA-style data. |
| Speech-LLaMA (Wu et al., 2023b) | Mel-Spectrogram(*C.*) → Text | X | CTC compressor + Transformer | Applies CTC compression to reduce the sequence length of the ended speech. |
| LLaSM (Shu et al., 2023) | Text + Whisper[†] (*C.*) → Text | X | Linear | An SLM that can follow instructions in both spoken and written forms. |
| BLSP (Wang et al., 2023a) | Text + Whisper[†] (*C.*) → Text | X | CNN | Demonstrates that a content-centric SLM can be trained using the response text of a text-based LLM, generated by conditioning on speech transcriptions. |
| SLM (Wang et al., 2023b) | Text + USM[†] (C.) → Text | X | Transformer | Demonstrates the capability of SLMs to generalize to unseen tasks while training only the modality adapter. Focuses on tasks related to speech content. |
| SALM (Chen et al., 2024d) | Text + Conformer Encoder[†] (C.) → Text | X | Conformer | Demonstrates the in-context learning capabilities of SLMs. Focuses on tasks related to speech content. |
| SALMONN (Tang et al., 2024) | Text + Whisper[†] (*C.*) + BEATS (*C.*) → Text | X | Window-level Q-Former | Demonstrates generalization to unseen tasks. Reports the issue of task overfitting. |
| COSMIC (Pan et al., 2024) | Text + Whisper[†] (*C.*) → Text | X | Window-level Q-Former | Instruction tuning on speech comprehension tests demonstrates unseen task generalization and in-context learning ability. |
| Qwen-Audio (Chu et al., 2023) | Text + Whisper[†] (*C.*) → Text | X | Pooling layer | Scales up training to diverse tasks and proposes a multitask training framework. |
| SpeechVerse (Das et al., 2024) | Text + Whisper[†] (*C.*) → Text | X | CNN | Scales up SLM training to diverse tasks and studies inference strategies to improve generalization to unseen tasks. |
| WavLLM (Hu et al., 2024) | Text + Whisper[†] (*C.*) + WavLM (*C.*) → Text | X | CNN + Bottleneck adapter + Linear | Combines two speech encoders to capture different aspects of the input speech. |
| DiscreteSLU (Shon et al., 2024) | Text + WavLM (*P.*) → Text | X | Embedding + CNN + Transformer + Linear | Proposes the use of discrete tokens for speech-aware text LMs. |
| BLSP-Emo (Wang et al., 2024a) | Text + Whisper[†] (*C.*) → Text | X | CNN | Extends BLSP for improved emotion understanding. Trains on LLM responses conditioned on speech transcripts + emotion tags. |
| DeSTA (Lu et al., 2024) | Text + Whisper[†] (*C.*) → Text | X | Q-Former | Proposes a descriptive speech-text alignment approach based on speech captioning. |
| Qwen2-Audio (Chu et al., 2024) | Text + Whisper[†] (*C.*) → Text | X | Pooling layer | Extends Qwen-Audio. Scales up the training dataset to 500k hours. Includes multi-task pre-training, supervised fine-tuning, and direct preference optimization. |
| DeSTA2 (Lu et al., 2025a) | Text + Whisper[†] (*C.*) → Text | X | Q-Former | Demonstrates that a versatile SLM can be trained using self-generated speech captions from a text LLM, conditioned on speech metadata, as its learning target. |
| DiVA (Held et al., 2025) | Text + Whisper[†] (*C.*) → Text | X | Q-Former | Trains a versatile SLM using self-generated speech captions from a text LLM, conditioned on speech transcriptions, via cross-modal token alignment loss and distillation from output embedding distance. |
| VoiceTextBlender (Peng et al., 2025b) | Text + Canary Encoder[†] (*C.*) → Text | X | Conformer | Proposes a joint speech-text supervised fine-tuning strategy to preserve the text LM's original performance. |
| Phi-4-Multimodal (Microsoft et al., 2025) | Text + Conformer Encoder(*C.*) → Text | X | MLP | A multimodal model that integrates text, vision, and speech/audio inputs using modality-specific routers to enable interference-free multimodal inference. |
| Audio-Reasoner (Xie et al., 2025) | Fine-tuned from Qwen2-Audio-Instruct | | | A large-scale audio language model for deep reasoning tasks on audio. |
| DeSTA2.5-Audio (Lu et al., 2025b) | Text + Whisper[†] (*C.*) → Text | X | Q-Former | Extends DeSTA2 with more training data. Finds that using captioning data generated by the text model that serves as the backbone of the SLM is critical. |
| Audio-Thinker (Wu et al., 2025c) | Fine-tuned from Qwen2-Audio-Instruct or Qwen2.5-Omni | | | A reinforcement learning approach to enhance the reasoning capabilities of SLMs. |

Table 6: A representative set of training and evaluation strategies for speech-aware text language models, along with key tasks, details on the training data, and key findings.

| Model | Training Strategy | Training Tasks | Evaluation Tasks | Training Data | Findings |
|---|---|---|---|---|---|
| SLM (Wang et al., 2023b) | IT | ASR, ST, Alpaca tasks | ASR, ST, contextual biasing, open-ended QA | multilingual ASR, ST datasets, Alpaca with TTS (93k hrs) | Fine-tuning only a lightweight adapter is sufficient for many tasks. |
| SALMONN (Tang et al., 2024) | Pre-training → IT | ASR, AAC, 15 audio and speech tasks | 8 seen audio and speech task + 7 unseen ST and SLU tasks | ASR, ST and SLU datasets (4k hrs) | Performs complex reasoning tasks like audio-based storytelling in zero-shot setting. |
| LTU-AS (Gong et al., 2023b) | 3-stage training | Classification, general QA | Audio and speech tasks, open-ended QA | Open-ASQA dataset (9.6M QA pairs) | Single instruction following model on both speech and audio tasks is feasible. |
| COSMIC (Pan et al., 2024) | IT | ASR, QA | ASR, SQA, ST, contextual biasing, ASR (out of domain) | TED-LIUM 3 (452 hours) | Shows few-shot in-context learning ability. |
| LTU (Gong et al., 2024) | 4-stage training | Classification + cesc., close-ended QA, general QA | Classification, captioning, open-ended QA | AQA (5M QA pairs) | Performs *open-ended* audio tasks. |
| SALM (Chen et al., 2024d) | IT | ASR, ST | ASR, ST, keyword boosting | LibriSpeech (960h), IWSLT 2023 (4.8k hrs) | Biases the model to predict keywords in the instruction. |
| Qwen-Audio (Chu et al., 2023) | Pre-training → IT | speech, audio and music tasks, audio dialogue | Speech, audio and music tasks, qualitative examples of audio analysis/editing | No details (30k hrs for ASR, >123k hrs in total) | Performs chat-based training to learn conversational ability. |
| DeSTA2 (Lu et al., 2025a) | IT | Audio captioning | Dynamic-SUPERB (Huang et al., 2024), AIR-Bench (Yang et al., 2024b) | Mixture of several datasets (155 hours) | Training only on audio captioning can generalize to other tasks. |
| DiscreteSLU (Shon et al., 2024) | IT | ASR, SQA, sentiment analysis, NER | WER, S2ST | Tedlium-3 (Hernandez et al., 2018) & SLUE (Shon et al., 2022) | Discrete speech token input can be competitive with continuous representations, in both seen and unseen speech domains, even in a zero-shot setting. |

