# OpenReview forum: "On The Landscape of Spoken Language Models: A Comprehensive Survey"
_TMLR — Accepted by TMLR_

### Review · Reviewer_BhXd · 2025-04-23

**Summary Of Contributions:**

This paper presents a comprehensive and well-structured survey of Spoken Language Models (SLMs), a rapidly growing area that seeks to extend the capabilities of large language models to speech. The authors propose a taxonomy of SLMs (pure speech LMs, speech+text LMs, and speech-aware text LMs) and systematically analyze their design choices, including tokenization strategies, modality adapters, sequence modeling architectures, and training procedures. The paper also reviews representative models in each category and introduces a functional definition of universal speech processing systems. Additionally, it identifies open challenges and outlines potential directions for future work. The survey synthesizes and unifies a fragmented body of literature, providing a clear reference point for ongoing and future research in this space.

**Audience:**

Yes

**Claims And Evidence:**

Yes

**Requested Changes:**

While Table 1 already provides a helpful typology of SLMs, the paper might benefit from a supplementary table or visual summary that focuses on model-level comparisons. This could include input/output modalities, types of training supervision, instruction-following capabilities, or the scope of evaluation. A qualitative overview of such dimensions could help readers quickly grasp the practical differences between models.

It could also be valuable to include a brief discussion of real-world deployment considerations, such as latency, memory efficiency, and interactive usability, which are particularly relevant for conversational or duplex SLMs.

To more clearly establish the paper’s contribution in relation to other recent surveys, the authors might consider including a brief comparative discussion. Highlighting differences in scope, taxonomy, or recency of coverage could help clarify how this paper complements or improves upon concurrent work.

For some briefly mentioned models, especially recent ones, providing slightly more descriptive context or explicit pointers to relevant sections or appendix tables would make the paper easier to navigate without requiring major expansion.

**Strengths And Weaknesses:**

The paper's main strength lies in its timeliness, clarity, and breadth of coverage. It introduces a coherent framework for understanding the landscape of SLMs, backed by clear terminology and structured analysis. The paper also succeeds in contextualizing recent developments within a broader trajectory from task-specific models to universal speech processing systems. Its writing is accessible, and the inclusion of architectural illustrations and classification tables greatly enhances comprehension.

However, the paper offers limited comparative synthesis across models, making it difficult to assess performance trade-offs or practical design implications. Some model descriptions, especially for newer or less well-documented systems, are brief and could benefit from more elaboration. Additionally, deployment concerns such as latency, inference cost, or real-time viability are largely omitted. While the survey distinguishes itself conceptually from prior work, its novelty relative to concurrent surveys could be more explicitly articulated.

---

> ### Author Response · Authors · 2025-07-08
> **Response to review**
>
> Thank you for your review and helpful suggestions!  We are in the process of revising the paper to address your and other reviewers’ points, as described below and in responses to other reviewers.
>
> 1.  **“... limited comparative synthesis across models, making it difficult to assess performance trade-offs or practical design implications”**
>
> We are revising Section 7 “Benchmarking and Evaluating SLMs” to include a summary of takeaways from existing benchmarks, which should help clarify the comparisons across models.  We want to point out, however, that benchmarking of SLMs is still not as thorough or consistent as we might like, as described in the “Evaluation” paragraph in Section 8 (“Challenges and Future Work”).  For this reason, the takeaways from current benchmarks are necessarily only partial.  One of the goals of our survey is to point out that evaluation is still an important gap in the literature.  Hopefully, including the key current benchmark results in our revision will help clarify the current state of this area.
>
>
> 2.  **“... the paper might benefit from a supplementary table or visual summary that focuses on model-level comparisons. This could include input/output modalities, types of training supervision, instruction-following capabilities, or the scope of evaluation.”**
>
> Table 4 in the Appendix includes this information for the category of speech-aware text LMs (the only category for which some of these dimensions are applicable).  Is there additional information you feel we should add?
>
> 3.  **“... include a brief discussion of real-world deployment considerations, such as latency, memory efficiency, and interactive usability”**
>
> This is a good point.  Unfortunately, at this point in the development of SLMs, the literature includes very limited information about these considerations.  This is an important gap, and we will add this point to Section 8 “Challenges and future work”.
>
> Even though there is a lack of existing literature discussing such practical points, we will still try our best to address your suggestion. For latency, in Section 7, we will report latency values and interactive usability metrics we have found in the literature (for example, in the new URO-Bench and Full-Duplex-Bench benchmarks). For efficiency, we will also summarise the model sizes along with benchmark performance of some SLM models in Section 7.
>
> 4.  **“To more clearly establish the paper’s contribution in relation to other recent surveys, the authors might consider including a brief comparative discussion.”**
>
> The paragraph “Related surveys” in the introduction describes the relationship to other surveys.  As mentioned in the discussion with Reviewer DySR, we plan to further clarify the relationship between our survey and Cui et al. 2024; specifically, we aim to provide a complementary survey that focuses on a unifying description and categorization of the field to help the reader navigate this landscape.  Besides this point, are there other points you feel are missing or unclear in the “Related surveys” paragraph?
>
> 5.  **“For some briefly mentioned models, especially recent ones, providing slightly more descriptive context or explicit pointers to relevant sections or appendix tables”**
>
> Thanks for the suggestion.  Could you please provide some examples of specific recent models and/or text lines/sections that you feel need more description or pointers?  As mentioned above, we plan to add key takeaways from recent benchmarks to help provide more context, and will also do another pass of proofreading to make sure that all needed pointers are included. In addition, as mentioned in the discussion with Reviewer DySR, we will also provide a short takeaway for each model in Table 3. We hope these efforts address your concern, but please let us know any specific spots you feel we should make sure to address.

---

### Review · Reviewer_DySR · 2025-05-04

**Summary Of Contributions:**

Following the success of text LLMs, there has been a large body of work trying to apply the same core principles (e.g. large scale pre-training; instruction tuning; etc) to speech processing. This survey paper organizes and summarizes papers in recent years on the topic of building more universal speech processing models, which this paper calls "spoken language models" (SLM).

Relevant models can be categorized based on a model's capability of ingesting and producing text/speech input (Table 1). This paper starts by outlining a functional definition of such SLMs (page 2):

> 1. It has both spoken input and spoken output with optional text input and/or output. The spoken input may serve as either an instruction or a context.
> 2. It is intended to be “universal”; that is, it should in principle be able to address arbitrary spoken language tasks, including both traditional tasks and more complex reasoning about spoken data.
> 3. It takes instructions or prompts in the form of natural language (either speech or text), and not, for example, only task specifiers (Radford et al., 2023) or soft prompts (Chang et al., 2024).

The rest of the paper then surveys the various important topics around SLMs, including
- Architecture components including speech encoders, modality adaptors, sequence models, and speech decoders (Sections 2 & 3)
- Pre- and post-training strategies (Section 4)
- Representative SLMs (Section 5)
- SLMs that can simultaneously listen and speak (duplex models, Section 6)
- Evaluation methods (Section 7)

Finally, Section 8 outlines the challenges and discusses possible directions of future work.

**Audience:**

Yes

**Broader Impact Concerns:**

None.

**Claims And Evidence:**

Yes

**Requested Changes:**

It would be greatly beneficial to readers of this paper if the weaknesses above can be better addressed.

**Strengths And Weaknesses:**

As mentioned in "Related Surveys" in page 3, there are a few contemporary survey papers on similar topics, in particular, [Cui et al, 2024](https://arxiv.org/pdf/2410.03751). Cui et al, 2024 and this paper are quite complementary in their strengths and weaknesses.

### Strengths (of this paper)

-   Comprehensiveness: While both papers cover the core of SLMs quite well, this paper covers the following aspects which are not sufficiently discussed in Cui et al, 2024:
    -   Modality adaptors
    -   Hierarchical token generation
    -   Interleaved speech/token generation

### Weaknesses (of this paper)

-   Presentation: While the information is all there in this paper, Cui et al, 2024 arguably does a better job in presenting the same information in a more efficiently digestable manner.
    -   The writing in this paper is often more verbose while Cui et al, 2024 is more succint.
    -   Contents in the Appendix are actually highly valuable for offering a presentation of more structured information, and thus should be incorporated into the flow of the main paper similar to how it's done in Cui et al, 2024.

-   Technical details: The descriptions of existing models and methods in this paper are often high level and abstract, and readers would find themselves knowing more technical details having read Cui et al, 2024. For example,
    -   Cui et al, 2024 starts by a convincing and detailed argument of why SLMs are better than traditional cascades. While the same discussion can be found in the bottom of page 2, it does not get the point through as strongly and clearly.
    -   Cui et al, 2024 is often better at clearly but succintly describing the individual contribution of important past work. While the same papers are also cited in this paper, they are usually only cited as part of the categorization effort.

---

> ### Author Response · Authors · 2025-07-08
> **Response to review**
>
> Thank you for your review and helpful suggestions!  We are in the process of revising the paper to address your and other reviewers’ points, as described below and in responses to other reviewers.
>
> In general, regarding comparisons with Cui et al. 2024:  We consider our survey to be complementary to theirs.  As you point out, there are certain aspects that we have chosen to cover in greater detail, while other aspects are summarized more briefly so as to highlight the important themes in the current research landscape.  We will clarify this in the “Related surveys” section in the introduction, and address your specific comments below.
>
> 1.  **“The writing in this paper is often more verbose while Cui et al, 2024 is more succint.”**
>
> We will proofread the paper again with an eye toward making the writing more succinct.
>
>
> 2.  **“Contents in the Appendix are actually highly valuable for offering a presentation of more structured information, and thus should be incorporated into the flow of the main paper”**
>
> We agree, and will move the timeline and key tables into the main text.
>
>
> 3.  **“Cui et al, 2024 starts by a convincing and detailed argument of why SLMs are better than traditional cascades. While the same discussion can be found in the bottom of page 2, it does not get the point through as strongly and clearly.”**
>
> We agree that the broad TMLR audience may benefit from a bit more detail in this discussion.  We will revise this section to include example tasks and settings where end-to-end SLMs have advantages over cascades.
>
> In addition, in Section 7 (the section on benchmarking), we will provide more results from existing benchmarks regarding the comparison between end-to-end SLMs and cascade models. We will point out those tasks for which today's state-of-the-art SLMs outperform cascade models, as well as those where cascade models still excel.
>
> 4.  **"Cui et al, 2024 is often better at clearly but succintly describing the individual contribution of important past work. While the same papers are also cited in this paper, they are usually only cited as part of the categorization effort."**
>
> Our survey takes a complementary approach to Cui et al. 2024.  While Cui et al. provide a more detailed digest of each paper they cover, our survey aims to give a unifying perspective that shows the key connections and distinctions between approaches in this area.
>
> That being said, we are happy to try to address your point without sacrificing this goal. We will restructure Table 3, adding a column that summarizes its key contribution or finding. This way, the main text of the overview paper can maintain a landscape-style perspective (as indicated by the title, we want to talk about "The Landscape of Spoken Language Models"), while readers can still refer to the specifics of each paper in Table 3. In fact, we have done something similar for some SLMs in Table 4 in the original paper. In addition, as mentioned above and in the reply to Reviewer VM9G, we plan to add some key takeaways from existing benchmarks.
>
> Does this address your concern?  If you feel there are specific papers we cite where we should further describe the individual contribution, in addition to the benchmark results and expansion of Table 3, please let us know, and we will take it into consideration in our revision.

---

### Review · Reviewer_VM9G · 2025-06-27

**Summary Of Contributions:**

This survey provides a thorough unification and overview of the field of Spoken Language Models (SLMs), categorizing them by architecture, training, evaluation, and challenges. It first gives a functional definition of SLM, identifying the discussion scope of this paper, and then makes in-depth discussions and analysis of the involved works. Finally, the achievements and limitations of existing methods are summarized, and then some research directions are provided for future works.

**Audience:**

Yes

**Broader Impact Concerns:**

No additional impact statement is required.

**Claims And Evidence:**

Yes

**Requested Changes:**

Overall, I consider it a high-quality survey, with a comprehensive overview and valuable insights. It could be better with the following minor changes:
1. If format permitted, the milestone timeline of Figure 6 in the appendix could be put at an earlier position in the main paper. It would make the survey more accessible, especially for newcomers.
2. Include comparison and analysis of quantitative results for key benchmarks to demonstrate the performance gaps.
3. Check and fix the typos.

**Strengths And Weaknesses:**

**Strengths**

- The survey covers a wide range of SLMs, including pure speech LMs, speech+text LMs, and speech-aware text LMs with clear categorizations. Meanwhile, it also captures recent advancements and benchmarks, making it highly relevant to current research.

- The general formulation of SLM is invaluable for understanding and unifying diverse architectures of the involved works, which is useful for building a standard of the field.

- The section of Challenges and future directions is well written, which figures out current critical problems in diverse aspects.

- The milestone timeline in the appendix is rich and comprehensive, which provides a clear development trajectory of the field for the readers.

**Weaknesses**
- The current version of the survey is a bit dense in text and not very easy to understand the scope at start, especially SLM is not a standard and widely known concept.

- While benchmarks are listed, there’s limited comparison of model performance. It's hard to learn the quantitative differences between the methods.

- Typos, e.g., "expemplified" in Figure 4 caption.

---

> ### Author Response · Authors · 2025-07-08
> **Response to review**
>
> Thank you for your review and helpful suggestions!  We are in the process of revising the paper to address your and other reviewers’ points, as described below and in responses to other reviewers.
>
>
> 1.   **“... the milestone timeline of Figure 6 in the appendix could be put at an earlier position in the main paper.”**
>
> We agree.  We will move Fig. 6 to Section 1.
>
>
> 2.  **Include comparison and analysis of quantitative results for key benchmarks to demonstrate the performance gaps.**
>
> We agree that it would be good to include more of the key takeaways from benchmark results.  We are revising Section 7 (“Benchmarking and Evaluating SLMs”) to include a summary of these takeaways, including performance results from widely-used benchmarks.
>
> We want to point out, however, that benchmarking of SLMs is still not as thorough or consistent as we might like, as described in the “Evaluation” paragraph in Section 8 (“Challenges and Future Work”).  For this reason, the takeaways from current benchmarks are necessarily only partial. One of the goals of our survey is to point out that evaluation is still an important gap in the literature.  Hopefully, including the key current benchmark results in our revision, as per your suggestion, will help clarify the current state of this area.
>
>
> 3.  **Typos**
>
> Thank you. We have fixed the typo you pointed out, and will proofread again for the next revision.

---

### Review · Reviewer_2ajH · 2025-07-12

**Summary Of Contributions:**

This paper presents a survey of Spoken Language Models (SLMs). The manuscript provides an overview of the field's history, provides a general framework for SLMs and covers overarching topics such as tokenization and construction. The paper then also discusses past and contemporary contributions to the field under new categorizations. Limitations and future directions are also discussed.

**Audience:**

Yes

**Claims And Evidence:**

Yes

**Requested Changes:**

Please see my weaknesses section for details. In addition, I think moving the timeline figure to the main body would be beneficial.

**Strengths And Weaknesses:**

Strengths:

- The paper is comprehensive in its coverage of the field.

- Informative and high quality figures help the flow of the paper and make digesting key concepts easy.

- The timeline figure is interesting and informative, the author’s should strongly consider moving this to the main body.

- Survey is timely and likely to be of interest to the audience.

Weaknesses:

- The author's make brief note of the closed-source nature of the field. I think expanding on this topic would be beneficial as it is somewhat atypical for machine learning and poses significant barriers to entry and progress compared to open-source domains.

- Discussion of benchmarks and evaluation methods is present, but discussion of relative performance of different SLMs/components is rather light.

- (Minor) Technical depth of the survey is fairly light when discussing existing works.

- (Minor) The paper could likely be a bit more concise.

---

> ### Author Response · Authors · 2025-07-13
> **Response to review**
>
> Thank you for your review and helpful suggestions!  We are in the process of revising the paper to address your and other reviewers’ points, as described below and in responses to other reviewers.
>
> *  **“The author's make brief note of the closed-source nature of the field. I think expanding on this topic would be beneficial as it is somewhat atypical for machine learning and poses significant barriers to entry and progress compared to open-source domains.”**
>
> As we mention in the “Open research” paragraph of Section 8 (“Challenges and future work”), we agree that this is unfortunate and an important issue for future work to address.  This situation is similar to that of language models for text, where open-source research has progressed but open-source models have lagged behind closed-source ones.  A similar evolution appears to be taking place in spoken language model research.  We will add this point about the relationship with text LM research, and hope that by highlighting this need we can encourage a faster evolution of open research.
>
> *  **Discussion of benchmarks and evaluation methods is present, but discussion of relative performance of different SLMs/components is rather light.**
>
> We agree that we can do a better job of discussing relative performance.  As mentioned in the discussion with other reviewers, we are revising Section 7 “Benchmarking and Evaluating SLMs” to include a summary of takeaways from existing benchmarks, which should help clarify the comparisons across models.  We want to point out, however, that benchmarking of SLMs is still not as thorough or consistent as we might like, as described in the “Evaluation” paragraph in Section 8 (“Challenges and Future Work”).  For this reason, the takeaways from current benchmarks are necessarily only partial.  One of the goals of our survey is to point out that evaluation is still an important gap in the literature.  Hopefully, including the key current benchmark results in our revision will help clarify the current state of this area.
>
> *  **(Minor) Technical depth of the survey is fairly light when discussing existing works.**
>
> We feel that this is to some extent a necessary tradeoff when attempting to unify the themes in a large body of research.  That being said, as mentioned in discussion with other reviewers, we plan to revise Table 3 to add key takeaways from each cited paper (without adding much to the main text, to avoid breaking the flow of the survey).  If you feel there is something else we should do to address your point, please let us know.
>
> *  **(Minor) The paper could likely be a bit more concise.**
>
> We agree that we could make the writing more concise in places, and will proofread the paper again to address this.
>
> *  **I think moving the timeline figure to the main body would be beneficial.**
>
> We agree and will make this change.

---

### Review · Reviewer_fvYi · 2025-07-17

**Summary Of Contributions:**

This paper presents a comprehensive survey of Spoken Language Models (SLMs), categorizing them into pure speech LMs, speech+text LMs, and speech-aware text LMs. It offers a unified architectural framework, reviews model components, training strategies, decoding methods, and highlights representative models. The work serves as a timely and structured reference for developing universal speech processing systems.

**Audience:**

Yes

**Claims And Evidence:**

Yes

**Requested Changes:**

Overall, this survey is logically structured and tells a coherent story, allowing readers to understand the latest developments and key challenges in the field of spoken language models (SLMs). To further improve the paper, the following changes are suggested:
-  Incorporate more critical analysis and comparison across models and strategies, highlighting their trade-offs, limitations, and suitable application scenarios.
- Simplify or summarise technically dense sections to make the content more accessible to readers less familiar with the field.

**Strengths And Weaknesses:**

Strengths
- Comprehensive coverage of models, training strategies, and applications in the SLM field.
- Clear taxonomy that effectively organizes the diverse approaches.
- Forward-looking insights into open challenges and future research directions.
- Well-structured presentation


Weaknesses
- Lack of critical comparison/analysis among models and strategies.

---

> ### Author Response · Authors · 2025-07-18
> **Response to review**
>
> Thank you for your review and helpful suggestions!  We are in the process of revising the paper to address your and other reviewers’ points, as described below and in responses to other reviewers.
>
> *  **Incorporate more critical analysis and comparison across models and strategies, highlighting their trade-offs, limitations, and suitable application scenarios.**
>
> We agree that we can do a better job of discussing relative performance.  As mentioned in the discussion with other reviewers, we are revising Section 7 “Benchmarking and Evaluating SLMs” to include a summary of takeaways from existing benchmarks, which should help clarify the comparisons across models.  We want to point out, however, that benchmarking of SLMs is still not as thorough or consistent as we might like, as described in the “Evaluation” paragraph in Section 8 (“Challenges and Future Work”).  For this reason, the takeaways from current benchmarks are necessarily only partial.  One of the goals of our survey is to point out that evaluation is still an important gap in the literature.  Hopefully, including the key current benchmark results in our revision will help clarify the current state of this area.  We also plan to revise Table 3 to add key takeaways from each cited paper (without adding much to the main text, to avoid breaking the flow of the survey).  If you feel there is something else we should do to address your point, please let us know.
>
> *  **Simplify or summarise technically dense sections to make the content more accessible to readers less familiar with the field.**
>
> We agree that we could make the writing more concise and readable in places, and will proofread the paper again to address this.

---

### Decision · Action_Editor_xvLq · 2025-09-03

**Recommendation:** Accept as is

**Audience:**

Yes

**Audience Explanation:**

reviewer fvYi said it "can serve as a timely and structured reference for developing universal speech processing systems."
reviewer 2ajH said that the "survey is timely and likely to be of interest to the audience."
reviewer VM9G said that the paper "contributes new insights in the unifying of the definition, method categorization, current limitations, and future works."
so i think it does have interest to the speech community and the tmlr audience

**Claims And Evidence:**

Yes

**Claims Explanation:**

every reviewer said yes to this question. reviewer BhXd said its main strength is its "timeliness, clarity, and breadth of coverage". reviewer VM9G stated that the "responses from the authors has addressed my concerns," after some rebuttals, so the paper's claims hold up under scrutiny and have been properly revised.